# DISENTANGLEMENT LEARNING VIA TOPOLOGY

## ABSTRACT

We propose TopDis (Topological Disentanglement), a method for learning disentangled representations via adding a multi-scale topological loss term. Disentanglement is a crucial property of data representations substantial for the explainability and robustness of deep learning models and a step towards high-level cognition. The state-of-the-art methods are based on VAE and encourage the joint distribution of latent variables to be factorized. We take a different perspective on disentanglement by analyzing topological properties of data manifolds. In particular, we optimize the topological similarity for data manifolds traversals. To the best of our knowledge, our paper is the first one to propose a differentiable topological loss for disentanglement learning. Our experiments have shown that the proposed TopDis loss improves disentanglement scores such as MIG, FactorVAE score, SAP score and DCI disentanglement score with respect to state-of-the-art results while preserving the reconstruction quality. Our method works in an unsupervised manner, permitting to apply it for problems without labeled factors of variation. The TopDis loss works even when factors of variation are correlated. Additionally, we show how to use the proposed topological loss to find disentangled directions in a trained GAN.

## 1 INTRODUCTION

Learning disentangled representations is a fundamental challenge in deep learning, as it has been widely recognized that achieving interpretable and robust representations is crucial for the success of machine learning models (Bengio et al., 2013). Disentangled representations, in which each component of the representation corresponds to one factor of variation (Desjardins et al., 2012; Bengio et al., 2013; Cohen & Welling, 2014; Kulkarni et al., 2015; Chen et al., 2016; Higgins et al., 2017; Tran et al., 2021; Feng et al., 2020; Gonzalez-Garcia et al., 2018), have been shown to be beneficial in a variety of areas within machine learning.

One key benefit of disentangled representations is that they enable effective domain adaptation, which refers to the ability of a model to generalize to new domains or tasks. Studies have shown that disentangled representations can improve performance in unsupervised domain adaptation (Yang et al., 2019; Peebles et al., 2020; Zou et al., 2020). Additionally, disentangled representations have been shown to be useful for zero-shot and few-shot learning, which are techniques for training models with limited labeled data (Bengio et al., 2013). Disentangled representations have also been shown to enable controllable image editing, which is the ability to manipulate specific aspects of an image while keeping the rest of the image unchanged (Wei et al., 2021; Wang & Ponce, 2021). This type of control can be useful in a variety of applications, such as image synthesis, style transfer and image manipulation. Furthermore, disentangled representations are also believed to be a vital component for achieving high-level cog-

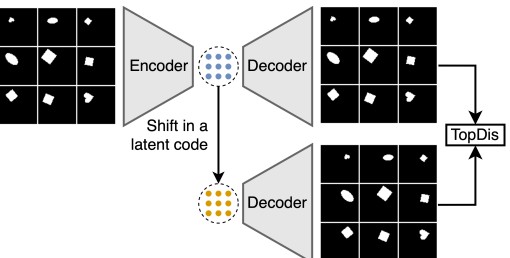

Figure 1: The TopDis pipeline process involves the following steps: encoding a batch of data samples, applying shift in a latent code, decoding both the original and the shifted latents, and finally calculating the TopDis regularization loss between the two resulting point clouds, for details see Section 4.

nition. High-level cognition refers to the ability of a model to understand and reason about the world, and disentangled representations can play a key role in achieving this goal (Bengio, 2018).

One line of research for finding disentangled representations is to modify the Variational Autoencoder (VAE) (Kingma & Welling, 2013) using some intuition, formalizing statistical independence of latent components (Higgins et al., 2017; Chen et al., 2018; Kim & Mnih, 2018), or the group theory based definition of disentanglement (Yang et al., 2021). Another line is to modify GANs (Goodfellow et al., 2014; Chen et al., 2016; Lin et al., 2020; Peebles et al., 2020; Wei et al., 2021) to enforce the change in a particular component being predictable or independent in some sense from other components.

At the same time, Locatello et al. (2019) stated the impossibility of fully unsupervised learning of disentangled representation with a statistical approach. But empirical evidence shows that disentanglement learning is possible, probably due to inductive bias either in the model or the dataset (Michlo et al., 2023; Rolinek et al., 2019). We follow Higgins et al. (2018), Section 3, where it is pointed out that one can achieve disentanglement w.r.t. the natural decomposition through active intervention, which in our case takes the form of the proposed group action shifts. Also our work is based on exploring topological properties of a data manifold. Thus, statistical arguments of Locatello et al. (2019) do not apply in our case.

In this paper, we take an approach to the problem of disentanglement learning. Our approach is grounded in the manifold hypothesis (Goodfellow et al., 2016) which posits that data points are concentrated in a vicinity of a low-dimensional manifold. For disentangled representations, it is crucial that the manifold has a specific property, namely, small topological dissimilarity between a point cloud given by a batch of data points and another point cloud obtained via the symmetry group action shift along a latent space axis. To estimate this topological dissimilarity, we utilize the tools from topological data analysis (Barannikov, 1994; Chazal & Michel, 2017). We then develop a technique for incorporating the gradient of this topological dissimilarity measure as into the training of VAE-type models.

Our contributions are the following:

- We propose TopDis (Topological Disentanglement), a method for unsupervised learning of disentangled representations via adding to a VAE-type loss the topological objective;
- Our approach uses group action shifts preserving the Gaussian distribution;
- We improve the reconstruction quality by applying gradient orthogonalization;
- Experiments show that the proposed topological regularization improves disentanglement metrics (MIG, FactorVAE score, SAP score, DCI disentanglement score) with respect to state-of-the-art results. Our methods works even if factors of generation are correlated.

## 2    RELATED WORK

In generative models, disentangled latent space can be obtained by designing specific architectures of neural networks (Karras et al., 2019) or optimizing additional loss functions. The latter approach can require true labels for factors of variation (Kulkarni et al., 2015; Kingma et al., 2014; Paige et al., 2017; Mathieu et al., 2016; Denton et al., 2017). However, the most interesting approach is to learn a disentangled latent space in an unsupervised manner. This is because not all data has labeled factors of variation, and at the same time, humans can easily extract factors of variation through their perception.

One of the most widely used generative models is the Variational Autoencoder (VAE) (Kingma & Welling, 2013). However, the VAE model alone is not able to achieve disentanglement. To address this limitation, researchers have proposed different variants of VAE such as $\beta$-VAE (Higgins et al., 2017), which aims to increase disentanglement by increasing the weight of KL divergence between the variational posterior and the prior. Increasing disentanglement in $\beta$-VAE often comes at the cost of a significant drop in reconstruction quality (Sikka et al., 2019). To overcome the trade-off between reconstruction and disentanglement, some researchers have proposed to use the concept of total correlation. In $\beta$-TCVAE (Chen et al., 2018), the KL divergence between the variational posterior and the prior is decomposed into three terms: index-code mutual information, total correlation (TC), and dimension-wise KL. The authors claim that TC is the most important term for learning

disentangled representations, and they penalize this term with an increased weight. However, they also note that it is difficult to estimate the three terms in the decomposition, and they propose a framework for training with the TC-decomposition using minibatches of data. The authors of FactorVAE (Kim & Mnih, 2018) propose to increase disentanglement also by reducing total correlation within latent factors. Instead of using the $\beta$-TCVAE approach, they rely on an additional discriminator which encourages the distribution of latent factors to be factorized and hence independent across the dimensions without significantly reducing the reconstruction loss.

In Locatello et al. (2019), the authors conduct a comprehensive empirical evaluation of a large amount of existing models for learning disentangled representations, taking into account the influence of hyperparameters and initializations. They find that the FactorVAE method achieves the best quality in terms of disentanglement and stability, while preserving the reconstruction quality of the generated images.

Approaches to interpretation of neural embeddings are developed in (Bertolini et al., 2022; Zhang et al., 2018; Zhou et al., 2018). Tools of topological data analysis were previously applied to disentanglement evaluation (Barannikov et al., 2022; Zhou et al., 2021).

## 3 BACKGROUND

### 3.1 VARIATIONAL AUTOENCODER

The Variational Autoencoder (VAE) (Kingma & Welling, 2013) is a generative model that encodes an object $x_n$ into a set of parameters of the posterior distribution $q_\phi(z|x_n)$, represented by an encoder with parameters $\phi$. Then it samples a latent representation from this distribution and decodes it into the distribution $p_\theta(x_n|z)$, represented by a decoder with parameters $\theta$. The prior distribution for the latent variables is denoted as $p(z)$. In this work, we consider the factorized Gaussian prior $p(z) = N(0, I)$, and the variational posterior for an observation is also assumed to be a factorized Gaussian distribution with the mean and variance produced by the encoder. The standard VAE model is trained by minimizing the negative Evidence Lower Bound (ELBO) averaged over the empirical distribution:

$$\mathcal{L}_{VAE} = \mathcal{L}_{rec} + \mathcal{L}_{KL} = \frac{1}{N} \sum_{n=1}^{N} \left[ -\mathbb{E}_q \left[\log p_\theta\left(x_n \mid z\right)\right] + \mathrm{KL}\left(q_\phi\left(z \mid x_n\right) \| p(z)\right) \right].$$

Several modifications of VAE for learning disentangled representations were proposed: $\beta$-VAE (Higgins et al., 2017), $\beta$-TCVAE (Chen et al., 2018), FactorVAE (Kim & Mnih, 2018), ControlVAE (Shao et al., 2020), DAVA (Estermann & Wattenhofer, 2023). The idea behind these methods to formalize statistical independence of latent components.

### 3.2 REPRESENTATION TOPOLOGY DIVERGENCE

Representation Topology Divergence (RTD) (Barannikov et al., 2022) is a topological tool comparing two point clouds $X, \tilde{X}$ with one-to-one correspondence between points. RTD compares multi-scale topological features together with their localization. The distances inside clouds $X, \tilde{X}$ define two weighted graphs $G^w, G^{\tilde{w}}$ with the same vertex set $X$, $w_{AB} = \mathrm{dist}(A, B)$, $\tilde{w}_{AB} = \mathrm{dist}(\tilde{A}, \tilde{B})$. For a threshold $\alpha$, the graphs $G^{w \leq \alpha}, G^{\tilde{w} \leq \alpha}$ are the $\alpha$-neighborhood graphs of $X$ and $\tilde{X}$. RTD tracks the differences in multi-scale topology between $G^{w \leq \alpha}, G^{\tilde{w} \leq \alpha}$ by comparing them with the graph $G^{\min(w, \tilde{w}) \leq \alpha}$, which contains an edge between vertices $A$ and $B$ iff an edge between $A$ and $B$ is present in either $G^{w \leq \alpha}$ or $G^{\tilde{w} \leq \alpha}$. Increasing $\alpha$ from 0 to the diameter of $X$, the connected components in $G^{w \leq \alpha}(X)$ change from $|X|$ separate vertices to one connected component with all vertices. Let $\alpha_1$ be the scale at which a pair of connected components $C_1, C_2$ of $G^{w \leq \alpha}$ becomes joined into one component in $G^{\min(w, \tilde{w}) \leq \alpha}$. Let at some $\alpha_2 > \alpha_1$, the components $C_1, C_2$ become also connected in $G^{w \leq \alpha}$. *R-Cross-Barcode*$_1(X, \tilde{X})$ is the multiset of intervals like $[\alpha_1, \alpha_2]$, see Figure 2. Longer intervals indicate in general the essential topological discrepancies between $X$ and $\tilde{X}$. By definition, RTD is the half-sum of intervals lengths in R-Cross-Barcode$_1(\tilde{X}, X)$ and R-Cross-Barcode$_1(X, \tilde{X})$. Formal definition of R-Cross-Barcode based on simplicial complexes is that it is the barcode of the graph $\hat{\mathcal{G}}^{w, \tilde{w}}$ from (Barannikov et al., 2022), see also Appendix N.

Figure 2 illustrates the calculation of RTD. The case with three clusters in $X$ merging into two clusters in $\tilde{X}$ is shown. Edges of $G^{\tilde{w}\leq\alpha}$ not in $G^{w\leq\alpha}$, are colored in orange. In this example there are exactly four edges of different weights $(13), (14), (23), (24)$ in the point clouds $X$ and $\tilde{X}$. The unique topological feature in *R-Cross-Barcode*$_1(X, \tilde{X})$ in this case is born at the threshold $\tilde{w}_{24}$ when the difference in the cluster structures of the two graphs arises, as the points 2 and 4 are in the same cluster at this threshold in $G^{\min(w,\tilde{w})\leq\alpha}$ and not in $G^{w\leq\alpha}$. This feature dies at the threshold $\alpha_2 = w_{23}$ since the clusters $(1, 2)$ and $(3, 4)$ are merged at this threshold in $G^{w\leq\alpha}$.

The differentiation of RTD is described in Trofimov et al. (2023), see also Appendix R.

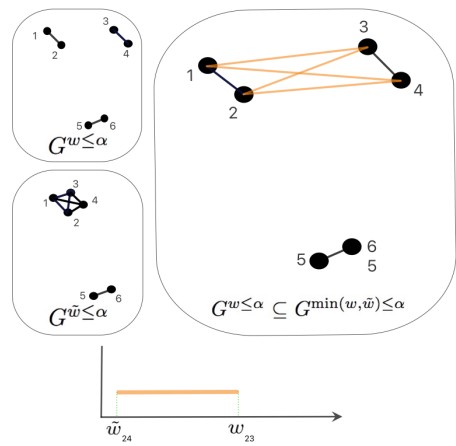

Figure 2: An example of RTD calculation.

## 4 METHOD

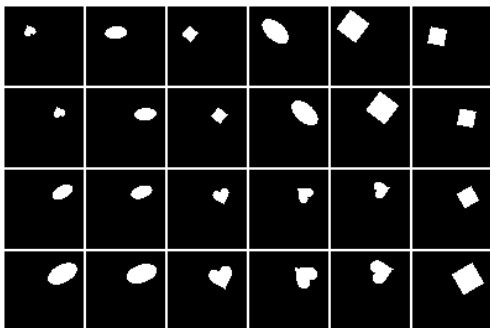

(a) Example of traversals in dSprites dataset.

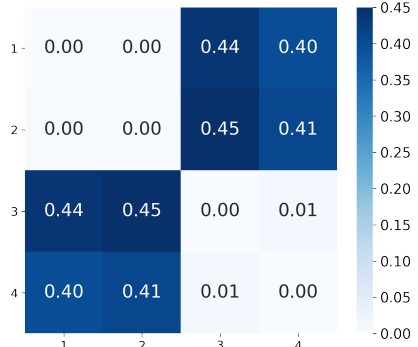

(b) RTD values beetween point clouds represented as rows in Figure 3a.

Figure 3: Left: rows represent point clouds (mini-batches). The 1st row represents some random batch of samples; the 2nd row is obtained by equally shifting samples from 1st row to the right; the 3rd row is placed the same as 2nd, but all objects are randomly transformed; the 4th row is a scaling of samples from 3rd row. The RTD value between 1st and 2nd point clouds is zero, as RTD between 3rd and 4th rows. While the RTD between 2nd and 3rd row have a large value because the topological structures of these two clouds are not similar.

### 4.1 TOPOLOGY-AWARE LOSS FOR GROUP ACTION

To illustrate our approach, we present an analysis of specific traversals in the dSprites dataset with known factors of variation. As shown in Figure 3a, we compute RTD values along shifts in latent space and demonstrate that transformations in disentangled directions have minimal topological dissimilarities between two sets of samples. In Figure 3b, the RTD values between point clouds represented as rows are displayed. As we explain below, the minimization of RTD is implied by continuity of the symmetry Lie group(oid) action on data distribution. Based on this, we focus on optimizing RTD as the measure of disentanglement in the form of TopDis regularization.

**Definition of VAE-based disentangled representation**. We propose that the desired outcome of VAE-based disentangled learning on data distribution $X$ consists of (cf. Higgins et al. (2018)):

1. The encoder $h : X \rightarrow Z$ and the decoder $f : Z \rightarrow X$ neural networks, $Z = \mathbb{R}^n$, maximizing ELBO, with the standard $N(0, I)$ prior distribution on $Z$.
2. Symmetry Lie group(oid) actions on distributions $X$ and $Z$, such that the decoder and the encoder are equivariant with respect to group(oid) action, $f(g(z)) = g(f(z)), h(g(x)) = g(h(x)), g \in G$.

3. A decomposition $G = G_1 \times \ldots \times G_n$, where $G_i \simeq \mathbb{R}$ are 1-parameter Lie subgroup(oid)s. We then distinguish two situations arising in examples: a) $G_i$ are commuting with each other, b) $G_i$ commutes with $G_j$ up to higher order $\mathcal{O}(C^2)$.

4. The Lie group(oid) $G$ action on the latent space decomposes and each $G_i$ acts only on a single latent variable $z_i$, preserving the prior $N(0,1)$ distribution on $z_i$; it follows from Proposition 4.1 that $G_i$ acts on $z_i$ via the shifts (1).

The concept of Lie group(oid) is a formalization of continuous symmetries, when a symmetry action is not necessarily applicable to all points. We gather necessary definitions in Appendix O.

The Lie group(oid) symmetry action by $g \in G$ on the support of data distribution is continuous and invertible. This implies that for any subset of the support of data distribution, the image of the subset under $g$ has the same homology or the same group of topological features. The preservation of topological features at multiple scales can be tested with the help of the representation topology divergence (RTD). If RTD is small between a sample from $X$ and its symmetry shift, then the groups of topological features at multiple scales are preserved.

Also the smallness of RTD implies the smallness of the disentanglement measure from (Zhou et al. (2021)) based on the geometry scores of data subsets conditioned to a fixed value of a latent code. Such subsets for different fixed values of the latent code are also related via the symmetry shift action, and if RTD between them is small, the distance between their persistence diagrams and hence the metric from loc cit is small as well.

## 4.2 GROUP ACTION SHIFTS PRESERVING THE GAUSSIAN DISTRIBUTION.

Our approach to learning disentangled representations is based on the use of an additional loss function that encourages the preservation of topological similarity in the generated samples when traversing along the latent space. Given a batch of data samples, $X = x_1, \ldots, x_N$, we sample the corresponding latent representations, $z_n \sim q_\phi(z|x_n)$, and the reconstructed samples, $\hat{x}_n \sim p_\theta(x|z_n)$. To ensure that the shifts in a latent code preserve the prior Gaussian distribution, we propose using the shifts defined by the equation:

$$z_{\text{shifted}} = F^{-1}(F(z \mid \rho, \sigma^2) + C \mid \rho, \sigma^2) \quad (1)$$

Shifts in the latent space are performed using the *cumulative* function $F(z \mid \rho, \sigma^2)$ of the Gaussian distribution. The mean value $\rho$ and variance $\sigma^2$ of the distribution are calculated as the empirical mean and variance of the latent code for the given sample of the data, see Algorithm 1.

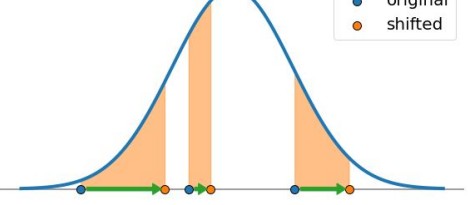

Figure 4: Shift of real line preserving $N(0,1)$, $C = 1/8$. The three orange curvilinear rectangles have the same area: $F(z_{\text{shifted}}) - F(z) = 1/8$

**Proposition 4.1.** *a) For any fixed $\rho, \sigma$, the equation (1) defines a local action of the additive group $\{C \mid C \in \mathbb{R}\}$ on real line. b) This abelian group(oid) action preserves the $N(\rho, \sigma^2)$ Gaussian distribution density. c) Conversely, if a local action of this abelian group preserves the $N(\rho, \sigma^2)$ distribution then the action is given by formula (1).*

See Appendix B for the proof and more details. This process is illustrated in Figure 4. Notice that, during the calculation of the topological term, we do not consider the data points with $F(z) + C > 1$ ($F(z) + C < 0$), i.e. whose latent codes are already at the very right (left) tail of the distribution and which thus cannot be shifted to the right (respectfully, left).

Let $q(z)$ be the aggregate posterior distribution over the latent space, aggregated over the whole dataset $X$. And let $q(z_i)$ be the similar aggregate distribution over the latent code $z_i$. The formula (1) is valid and defines symmetry shifts if we replace the standard normal distribution by any distribution over the real line, we use it with the distribution $q(z_i)$ over the $i-$th latent codes.

**Proposition 4.2.** *a) If the distribution $q(z)$ is factorized into product $q(z) = \prod_i q(z_i)$, then the shift defined by the formula (1) and acting on a single latent code $z_i$ and leaving other latent codes fixed, preserves the latent space distribution $q(z)$. This defines the $G_i$ groupoid action on $z$ for any $i$, whose action is then extended to points of the initial dataset $X$ with the help of the decoder-encoder.*

---

**Algorithm 1** Latent traversal with a shift in the latent space.

---

**Input:** $z \in \mathbb{R}^{N \times d}$ – an array of latent representations from encoder. $C$ – the shift value. $F(z \mid \rho, \sigma^2)$ – the cumulative function for $\mathcal{N}(\rho, \sigma^2)$ distribution.
   $i \sim \{1, \ldots, d\}$, random choice of latent code
   $s \sim \{-C, C\}$, random choice of shift direction.
   $\rho \leftarrow \mathrm{mean}(z^{(i)})$, empirical mean value for $i$-th latent representation along batch.
   $\sigma^2 \leftarrow \mathrm{var}(z^{(i)})$, empirical variance for the $i$-th latent representation along batch.
   $p \leftarrow F(z^{(i)} \mid \rho, \sigma^2)$, p-values of batch along $i$-th latent code, $p \in \mathbb{R}^N$
   $\mathcal{J} = \{j \mid p_j + s \in (0, 1)\}$, valid set of the batch elements that can be shifted
   $z_{\mathrm{original}} \leftarrow \{z_j \mid j \in \mathcal{J}\}$, batch of valid original latents
   $z_{\mathrm{shifted}}^{(i')} \leftarrow z_{\mathrm{original}}^{(i')}$, copy of latents $z_{\mathrm{original}}^{(i')}$, $i' \neq i$
   $z_{\mathrm{shifted}}^{(i)} \leftarrow \{F^{-1}(p_j + s \mid \rho, \sigma^2) \mid j \in \mathcal{J}\}$, apply the shift only along the $i$-th latent code.
**Return:** $z_{\mathrm{original}}, \ z_{\mathrm{shifted}}$ – valid original and shifted latents. $z_{\mathrm{original}}, \ z_{\mathrm{shifted}} \in \mathbb{R}^{|\mathcal{J}| \times d}$

---

**Algorithm 2** The TopDis loss.

---

**Input:** $X \in \mathbb{R}^{N \times C \times H \times W}$, VAE parameters $\phi, \theta$, $p \in \{1, 2\}$ – an exponent, $C$ – the shift scale.
   $\mu_z, \sigma_z^2 \leftarrow q_\phi(z|X)$, posterior parameters from encoder given batch $X$.
   $z_{\mathrm{original}}, \ z_{\mathrm{shifted}}$ – valid original and shifted latents, obtained by Algorithm 1
   $\hat{X}_{\mathrm{original}} \sim p_\theta(x|z_{\mathrm{original}})$, a reconstruction of initial batch $X$
   $\hat{X}_{\mathrm{shifted}} \sim p_\theta(x|z_{\mathrm{shifted}})$, a generation of modified $X$ after applying shift along some fixed latent code.
   $\mathcal{L}_{\mathcal{TD}} \leftarrow \mathrm{RTD}^{(p)}(\hat{X}_{\mathrm{original}}, \hat{X}_{\mathrm{shifted}})$
**Return:** $\mathcal{L}_{\mathcal{TD}}$ – topological loss term.

---

> b) *Conversely, if $q(z)$ is preserved for any $i$ by the shifts acting on $z_i$ and defined via formula (1) from the distribution $q(z_i)$, then $q(z) = \prod_i q(z_i)$.*
>
> The proof is given in Appendix C

## 4.3 THE TOPDIS LOSS

The TopDis regularization loss is calculated using the Representation Topology Divergence (RTD) measure, which quantifies the dissimilarity between two point clouds with one-to-one correspondence. The reconstructed batch of images, $\hat{X}$, is considered as a point cloud in the $\mathbb{R}^{H \times W \times C}$ space [1], $H$, $W$, and $C$ are the height, width, and number of channels of the images respectively. The one-to-one correspondence between the original and shifted samples is realized naturally by the shift in the latent space. Finally, having an original and shifted point clouds:

$$\hat{X}_{\mathrm{original}} \sim p_\theta(x|z_{\mathrm{original}}), \ \hat{X}_{\mathrm{shifted}} \sim p_\theta(x|z_{\mathrm{shifted}}), \tag{2}$$

we propose the following topological regularization term (Algorithm 2):

$$\mathcal{L}_{TD} = \mathrm{RTD}^{(p)}(\hat{X}_{\mathrm{original}}, \hat{X}_{\mathrm{shifted}}), \tag{3}$$

where the superscript $(p)$ in $\mathrm{RTD}^{(p)}$ stands for using sum of the lengths of intervals in R-Cross-Barcode$_1$ to the $p-$th power. The $\mathcal{L}_{TD}$ term imposes a penalty for data point clouds having different topological structures, like the 2nd and the 3rd rows in Figure 3a. Both standard values $p = 1$ and $p = 2$ perform well. In some circumstances, the $p = 2$ value is more appropriate because it penalizes more the significant variations in topology structures.

In this work, we propose to use the topological regularization term $\mathcal{L}_{TD}$, in addition to the VAE-based loss:

$$\mathcal{L} = \mathcal{L}_{VAE-based} + \gamma \mathcal{L}_{TD}. \tag{4}$$

---

[1] For complex images, RTD and the TopDis loss can be calculated in a representation space instead of the pixel space $X$.

All variants of VAEs ($\beta$-VAE, FactorVAE, ControlVAE, DAVA) are modified accordingly. The computational complexity of $\mathcal{L}_{TD}$ is discussed in Appendix M. We analyze sensitivity of the proposed approach on the value of $\gamma$ in (4) in Appendix Q.

## 4.4 GRADIENT ORTHOGONALIZATION

As all regularization terms, the $\mathcal{L}_{TD}$ minimization may lead to lack of reconstruction quality. In order to achieve state-of-the-art results while minimizing the topological regularization term $\mathcal{L}_{TD}$, we apply the gradient orthogonalization between $\mathcal{L}_{TD}$ and the reconstruction loss term $\mathcal{L}_{rec}$. Specifically, if the scalar product between $\nabla_{\phi,\theta}\mathcal{L}_{rec}$ and $\nabla_{\phi,\theta}\mathcal{L}_{TD}$ is negative, then we adjust the gradients from our $\mathcal{L}_{TD}$ loss to be orthogonal to those from $\mathcal{L}_{rec}$ by applying the appropriate linear transformation:

$$\nabla^{ort}\mathcal{L}_{TD} = \nabla\mathcal{L}_{TD} - \frac{\langle\nabla\mathcal{L}_{TD}, \nabla\mathcal{L}_{rec}\rangle}{\langle\nabla\mathcal{L}_{rec}, \nabla\mathcal{L}_{rec}\rangle}\nabla\mathcal{L}_{rec}. \tag{5}$$

This technique helps to maintain a balance between the reconstruction quality and the topological regularization, thus resulting in improved overall performance. We provide an ablation study of gradient orthogonalization technique in Appendix P.

## 5 EXPERIMENTS

### 5.1 EXPERIMENTS ON STANDARD BENCHMARKS

In the experimental section of our work, we evaluate the effectiveness of the proposed TopDis regularization technique. Specifically, we conduct a thorough analysis of the ability of our method to learn disentangled latent spaces using various datasets and evaluation metrics. We compare the results obtained by our method with the state-of-the-art models and demonstrate the advantage of our approach in terms of disentanglement and preserving reconstruction quality.

**Datasets**. We used popular benchmarks: dSprites (Matthey et al., 2017), 3D Shapes (Burgess & Kim, 2018), 3D Faces (Paysan et al., 2009), MPI 3D (Gondal et al., 2019), CelebA (Liu et al., 2015). See description of the datasets in Appendix J. Although the datasets dSprites, 3D Shapes, 3D Faces are synthetic, the known true factors of variation allow accurate supervised evaluation of disentanglement. Hence, these datasets are commonly used in both classical and most recent works on disentanglement (Burgess et al., 2017; Kim & Mnih, 2018; Estermann & Wattenhofer, 2023; Roth et al., 2022). Finally, we examine the real-life setup with the CelebA (Liu et al., 2015) dataset.

**Methods**. We combine the TopDis regularizer with the FactorVAE (Kim & Mnih, 2018), $\beta$-VAE (Higgins et al., 2017), ControlVAE (Shao et al., 2020), DAVA (Estermann & Wattenhofer, 2023). Also, we provide separate comparisons with $\beta$-TCVAE (Chen et al., 2018) and vanilla VAE (Kingma & Welling, 2013). Following the previous work Kim & Mnih (2018), we used similar architectures for the encoder, decoder and discriminator (see Appendix D), the same for all models. The hyperparameters and other training details are in Appendix L. We set the latent space dimensionality to 10. Since the quality of disentanglement has high variance w.r.t. network initialization (Locatello et al., 2019), we conducted multiple runs of our experiments using different initialization seeds[2] and averaged results.

**Evaluation**. Not all existing metrics were shown to be equally useful and suitable for disentanglement (Dittadi et al., 2021), (Locatello et al., 2019). Due to this, hyperparameter tuning and model selection may become controversial. Moreover, in the work Carbonneau et al. (2022), the authors conclude that the most appropriate metric is DCI disentanglement score (Eastwood & Williams, 2018), the conclusion which coincides with another line of research Roth et al. (2022). Based on the existing results about metrics' applicability, we restricted evaluation to measuring the following disentanglement metrics: the Mutual Information Gap (MIG) (Chen et al., 2018), the FactorVAE score (Kim & Mnih, 2018), DCI disentanglement score, and Separated Attribute Predictability (SAP) score (Kumar et al., 2017). Besides its popularity, these metrics cover all main approaches to evaluate the disentanglement of generative models (Zaidi et al., 2020): information-based (MIG), predictor-based (SAP score, DCI disentanglement score), and intervention-based (FactorVAE score).

---

[2]see Appendix K for more details.

Table 1: Evaluation on the benchmark datasets. **Bold** denotes the best variant in the pair with vs. without the TopDis loss. Blue denotes the best method for a dataset/metric.

| Method | FactorVAE score | MIG | SAP | DCI, dis. |
|---|---|---|---|---|
| dSprites | | | | |
| $\beta$-TCVAE | $0.810 \pm 0.058$ | $0.332 \pm 0.029$ | $0.045 \pm 0.004$ | $0.543 \pm 0.049$ |
| $\beta$-TCVAE + TopDis (ours) | $\mathbf{0.821 \pm 0.034}$ | $\mathbf{0.341 \pm 0.021}$ | $\mathbf{0.051 \pm 0.004}$ | $\mathbf{0.556 \pm 0.042}$ |
| $\beta$-VAE | $0.807 \pm 0.037$ | $0.272 \pm 0.101$ | $0.065 \pm 0.002$ | $0.440 \pm 0.102$ |
| $\beta$-VAE + TopDis (ours) | $\color{blue}\mathbf{0.833 \pm 0.016}$ | $\mathbf{0.348 \pm 0.028}$ | $\mathbf{0.066 \pm 0.015}$ | $\mathbf{0.506 \pm 0.050}$ |
| ControlVAE | $0.806 \pm 0.012$ | $0.333 \pm 0.037$ | $0.056 \pm 0.002$ | $0.557 \pm 0.009$ |
| ControlVAE + TopDis (ours) | $\mathbf{0.810 \pm 0.012}$ | $\mathbf{0.344 \pm 0.029}$ | $\mathbf{0.059 \pm 0.002}$ | $\color{blue}\mathbf{0.578 \pm 0.007}$ |
| FactorVAE | $0.819 \pm 0.028$ | $0.295 \pm 0.049$ | $0.053 \pm 0.006$ | $\mathbf{0.534 \pm 0.029}$ |
| FactorVAE + TopDis (ours) | $\mathbf{0.824 \pm 0.038}$ | $\color{blue}\mathbf{0.356 \pm 0.025}$ | $\color{blue}\mathbf{0.082 \pm 0.001}$ | $0.521 \pm 0.044$ |
| DAVA | $0.746 \pm 0.099$ | $0.253 \pm 0.058$ | $0.024 \pm 0.015$ | $0.395 \pm 0.054$ |
| DAVA + TopDis (ours) | $\mathbf{0.807 \pm 0.010}$ | $\mathbf{0.344 \pm 0.010}$ | $\mathbf{0.048 \pm 0.012}$ | $\mathbf{0.551 \pm 0.019}$ |
| 3D Shapes | | | | |
| $\beta$-TCVAE | $0.909 \pm 0.079$ | $0.693 \pm 0.053$ | $0.113 \pm 0.070$ | $0.877 \pm 0.018$ |
| $\beta$-TCVAE + TopDis (ours) | $\mathbf{1.0 \pm 0.0}$ | $\mathbf{0.751 \pm 0.051}$ | $\mathbf{0.147 \pm 0.064}$ | $\mathbf{0.901 \pm 0.014}$ |
| $\beta$-VAE | $0.965 \pm 0.060$ | $0.740 \pm 0.141$ | $0.143 \pm 0.071$ | $0.913 \pm 0.147$ |
| $\beta$-VAE + TopDis (ours) | $\color{blue}\mathbf{1.0 \pm 0.0}$ | $\color{blue}\mathbf{0.839 \pm 0.077}$ | $\color{blue}\mathbf{0.195 \pm 0.030}$ | $\color{blue}\mathbf{0.998 \pm 0.004}$ |
| ControlVAE | $0.746 \pm 0.094$ | $0.433 \pm 0.094$ | $0.091 \pm 0.068$ | $0.633 \pm 0.093$ |
| ControlVAE + TopDis (ours) | $\mathbf{0.806 \pm 0.046}$ | $\mathbf{0.591 \pm 0.055}$ | $\mathbf{0.125 \pm 0.02}$ | $\mathbf{0.795 \pm 0.098}$ |
| FactorVAE | $0.934 \pm 0.058$ | $0.698 \pm 0.151$ | $0.099 \pm 0.064$ | $0.848 \pm 0.129$ |
| FactorVAE + TopDis (ours) | $\mathbf{0.975 \pm 0.044}$ | $\mathbf{0.779 \pm 0.036}$ | $\mathbf{0.159 \pm 0.032}$ | $\mathbf{0.940 \pm 0.089}$ |
| DAVA | $0.800 \pm 0.095$ | $0.625 \pm 0.061$ | $0.099 \pm 0.016$ | $0.762 \pm 0.088$ |
| DAVA + TopDis (ours) | $\mathbf{0.847 \pm 0.092}$ | $\mathbf{0.679 \pm 0.112}$ | $\mathbf{0.010 \pm 0.043}$ | $\mathbf{0.836 \pm 0.074}$ |
| 3D Faces | | | | |
| $\beta$-TCVAE | $\color{blue}1.0 \pm 0.0$ | $0.568 \pm 0.063$ | $0.060 \pm 0.017$ | $0.822 \pm 0.033$ |
| $\beta$-TCVAE + TopDis (ours) | $\color{blue}1.0 \pm 0.0$ | $\mathbf{0.591 \pm 0.058}$ | $\mathbf{0.062 \pm 0.011}$ | $\mathbf{0.859 \pm 0.031}$ |
| $\beta$-VAE | $\color{blue}1.0 \pm 0.0$ | $\mathbf{0.561 \pm 0.017}$ | $\mathbf{0.058 \pm 0.008}$ | $\color{blue}\mathbf{0.873 \pm 0.018}$ |
| $\beta$-VAE + TopDis (ours) | $\color{blue}1.0 \pm 0.0$ | $0.545 \pm 0.005$ | $0.052 \pm 0.004$ | $0.854 \pm 0.013$ |
| ControlVAE | $\color{blue}1.0 \pm 0.0$ | $0.447 \pm 0.011$ | $0.058 \pm 0.008$ | $0.713 \pm 0.007$ |
| ControlVAE + TopDis (ours) | $\color{blue}1.0 \pm 0.0$ | $\mathbf{0.477 \pm 0.004}$ | $\color{blue}\mathbf{0.074 \pm 0.007}$ | $\mathbf{0.760 \pm 0.014}$ |
| FactorVAE | $\color{blue}1.0 \pm 0.0$ | $0.593 \pm 0.058$ | $0.061 \pm 0.014$ | $0.848 \pm 0.011$ |
| FactorVAE + TopDis (ours) | $\color{blue}1.0 \pm 0.0$ | $\color{blue}\mathbf{0.626 \pm 0.026}$ | $\mathbf{0.062 \pm 0.013}$ | $\mathbf{0.867 \pm 0.037}$ |
| DAVA | $\color{blue}1.0 \pm 0.0$ | $0.527 \pm 0.002$ | $0.047 \pm 0.009$ | $\mathbf{0.822 \pm 0.006}$ |
| DAVA + TopDis (ours) | $\color{blue}1.0 \pm 0.0$ | $\mathbf{0.536 \pm 0.012}$ | $\mathbf{0.052 \pm 0.011}$ | $0.814 \pm 0.008$ |
| MPI 3D | | | | |
| $\beta$-TCVAE | $0.365 \pm 0.042$ | $0.174 \pm 0.018$ | $0.080 \pm 0.013$ | $0.225 \pm 0.061$ |
| $\beta$-TCVAE + TopDis (ours) | $\mathbf{0.496 \pm 0.039}$ | $\mathbf{0.280 \pm 0.013}$ | $\mathbf{0.143 \pm 0.009}$ | $\mathbf{0.340 \pm 0.055}$ |
| $\beta$-VAE | $0.442 \pm 0.063$ | $0.250 \pm 0.063$ | $0.104 \pm 0.028$ | $0.255 \pm 0.032$ |
| $\beta$-VAE + TopDis (ours) | $\mathbf{0.487 \pm 0.040}$ | $\mathbf{0.363 \pm 0.028}$ | $\mathbf{0.181 \pm 0.030}$ | $\mathbf{0.348 \pm 0.027}$ |
| ControlVAE | $0.394 \pm 0.020$ | $0.165 \pm 0.053$ | $0.107 \pm 0.004$ | $0.183 \pm 0.031$ |
| ControlVAE + TopDis (ours) | $\mathbf{0.545 \pm 0.031}$ | $\mathbf{0.225 \pm 0.019}$ | $\mathbf{0.154 \pm 0.005}$ | $\mathbf{0.251 \pm 0.027}$ |
| FactorVAE | $0.628 \pm 0.039$ | $0.351 \pm 0.034$ | $0.209 \pm 0.043$ | $0.414 \pm 0.031$ |
| FactorVAE + TopDis (ours) | $\color{blue}\mathbf{0.677 \pm 0.017}$ | $\color{blue}\mathbf{0.420 \pm 0.033}$ | $\color{blue}\mathbf{0.254 \pm 0.031}$ | $\color{blue}\mathbf{0.469 \pm 0.034}$ |
| DAVA | $0.407 \pm 0.023$ | $0.129 \pm 0.028$ | $0.076 \pm 0.034$ | $0.227 \pm 0.035$ |
| DAVA + TopDis (ours) | $\mathbf{0.613 \pm 0.056}$ | $\mathbf{0.289 \pm 0.009}$ | $\mathbf{0.148 \pm 0.019}$ | $\mathbf{0.391 \pm 0.023}$ |

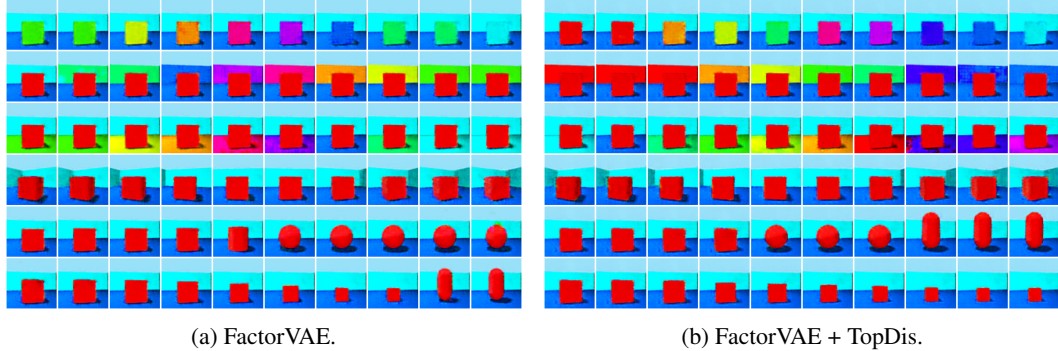

(a) FactorVAE.             (b) FactorVAE + TopDis.

Figure 5: FactorVAE and FactorVAE + TopDis latent traversals on 3D Shapes.

### 5.1.1 QUANTITATIVE EVALUATION

The results presented in Table 1 demonstrate that TopDis regularized models outperform the original ones for all datasets and almost all quality measures. The addition of the TopDis regularizer improves the results as evaluated by FactorVAE score, MIG, SAP, DCI: on dSprites up to 8%, 35%, 100%, 39%, on 3D Shapes up to 8%, 36%, 60%, 25%, on 3D Faces up to 0%, 6%, 27%, 6% and up to 50%, 124%, 94%, 72% on MPI 3D respectively across all models. The best variant for a dataset/metrics is almost always a variant with the TopDis loss, in 94% cases. In addition, our approach preserves the reconstruction quality, see Table 4 in Appendix E.

### 5.1.2 QUALITATIVE EVALUATION

In order to qualitatively evaluate the ability of our proposed TopDis regularization to learn disentangled latent representations, we plot the traversals along a subset of latent codes that exhibit the most significant changes in an image. As a measure of disentanglement, it is desirable for each latent code to produce a single factor of variation. We compare traversals from FactorVAE and FactorVAE+TopDis decoders. The corresponding Figures 18, 19 are in Appendix V.

**dSprites.** Figures 18a and 18b show that the TopDis regularizer helps to outperform simple FactorVAE in terms of visual perception. The simple FactorVAE model is observed to have entangled rotation and shift along axes (raws 1,2,5 in Figure 18a), even though the Total Correlation in both models is minimal, which demonstrates the impact of the proposed regularization method.

**3D Shapes.** Figures 5a and 5b and show that our TopDis regularization leads to the superior disentanglement of the factors as compared to the simple FactorVAE model, where the shape and the scale factors remain entangled in the last row (see Figure 5a).

**3D Faces.** FactorVAE+TopDis (Figure 18f) outperforms FactorVAE (Figure 18e) in terms of disentangling the main factors such as azimuth, elevation, and lighting from facial identity. On top of these figures we highlight the azimuth traversal. The advantage of TopDis is seen from the observed preservations in facial attributes such as the chin, nose, and eyes.

**MPI 3D.** Here, the entanglement between the size and elevation factors is particularly evident when comparing the bottom two rows of Figures 18g and 18h. In contrast to the base FactorVAE, which left these factors entangled, our TopDis method successfully disentangles them.

**CelebA.** For this dataset, we show the most significant improvements obtained by adding the TopDis loss in Figure 19. The TopDis loss improves disentanglement of skin tone and lightning compared to basic FactorVAE, where these factor are entangled with other factors - background and hairstyle.

### 5.2 LEARNING DISENTANGLED REPRESENTATIONS FROM CORRELATED DATA

Existing methods for disentanglement learning make unrealistic assumptions about statistical independence of factors of variations (Träuble et al., 2021). Synthetic datasets (dSprites, 3D Shapes, 3D Faces, MPI 3D) also share this assumption. However, in real world, causal factors are typically correlated. We carry out a series of experiments with shared confounders (one factor correlated to all others, (Roth et al., 2022)). The TopDis loss isn't based on assumptions of statistical independence. Addition of the TopDis loss gives a consistent improvement in all quality measures in this setting, see Table 10 in Appendix U.

### 5.3 UNSUPERVISED DISCOVERY OF DISENTANGLED DIRECTIONS IN STYLEGAN

We perform additional experiments to study the ability of the proposed topology-based loss to infer disentangled directions in a pre-trained StyleGAN (Karras et al., 2019). We searched for disentangled directions within the space of principal components in latent space by optimizing the multi-scale topological difference after a shift along this axis $\mathrm{RTD}(\hat{X}_{\mathrm{original}}, \hat{X}_{\mathrm{shifted}})$. We were able to find three disentangled directions: azimuth, smile, hair color. See Figure 6 and Appendix I for more details. Comparison of methods dedicated to the unsupervised discovery of disentangled directions in StyleGAN is qualitative

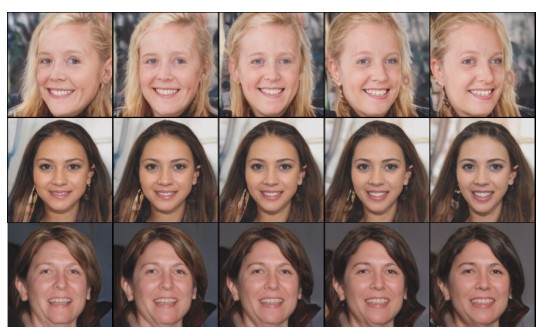

Figure 6: Three disentangled directions discovered by TopDis in StyleGAN: azimuth, smile, hair color.

since the FFHQ dataset doesn't have labels. We do not claim that our method outperforms alternatives (Härkönen et al., 2020), as our goal is rather to demonstrate the applicability of the TopDis loss for this problem.

## 6 CONCLUSION

Our method, the Topological Disentanglement, has demonstrated its effectiveness in learning disentangled representations, in an unsupervised manner. The experiments on the dSprites, 3D Shapes, 3D Faces and MPI 3D datasets have shown that an addition of our TopDis regularizer improves $\beta$-VAE, ControlVAE, FactorVAE and DAVA models in terms of disentanglement scores (MIG, FactorVAE score, SAP score, DCI disentanglement score) while preserving the reconstruction quality. Inside our method, there is the idea of applying the topological dissimilarity to optimize disentanglement that can be added to any existing approach or used alone. We proposed to apply group action shifts preserving the Gaussian distribution in the latent space. To preserve the reconstruction quality, the gradient orthogonalization were used. Our method isn't based on the statistical independence assumption and brings improvement of quality measures even if factors of variation are correlated. In this paper, we limited ourselves with the image domain for easy visualization of disentangled directions. Extension to other domains (robotics, time series, etc.) is an interesting avenue for further research.

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

# A    RANDOM DATASET SAMPLES

In Figures 7, 8, 9, 10 and 11 we demonstrate random samples from dSprites, 3D Shapes, 3D Faces, CelebA and MPI 3D datasets respectively.

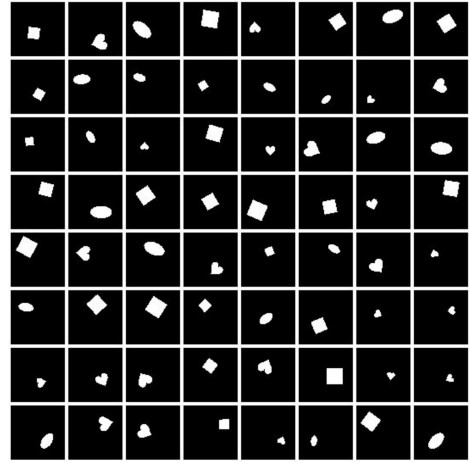

Figure 7: dSprites ($64 \times 64$)

Figure 8: 3D Shapes ($64 \times 64$)

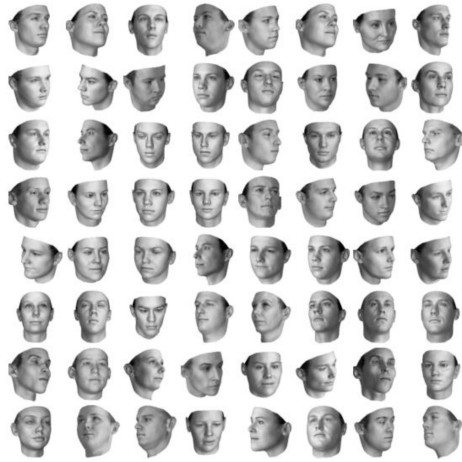

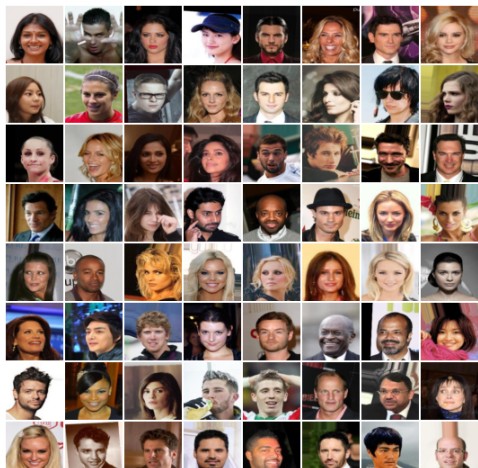

Figure 9: 3D Faces ($64 \times 64$)

Figure 10: CelebA ($64 \times 64$)

# B    PROOF OF PROPOSITION 4.1

a) Two consecutive shifts defined in 1 give

$$F^{-1}(F(F^{-1}(F(z \mid \rho, \sigma^2) + C_1 \mid \rho, \sigma^2) + C_2 \mid \rho, \sigma^2) = F^{-1}(F(z \mid \rho, \sigma^2) + C_1 + C_2 \mid \rho, \sigma^2)$$

So the two consecutive shifts with $C_1, C_2$ is the same as the single shift with $C_1 + C_2$.

b) We have for a given shift with parameter $C$ and any pair of shifted points $z_{\text{shifted}}, \tilde{z}_{\text{shifted}} \in \mathbb{R}$ :

$$F(\tilde{z}_{\text{shifted}}) - F(z_{\text{shifted}}) = (F(\tilde{z}) + C) - (F(z) + C) = F(\tilde{z}) - F(z) \tag{6}$$

i.e. if the shift of points $z, \tilde{z} \in \mathbb{R}$ is defined, then the $N(\rho, \sigma^2)$ measure of the line segment $[z, \tilde{z}]$ is preserved under the shift.

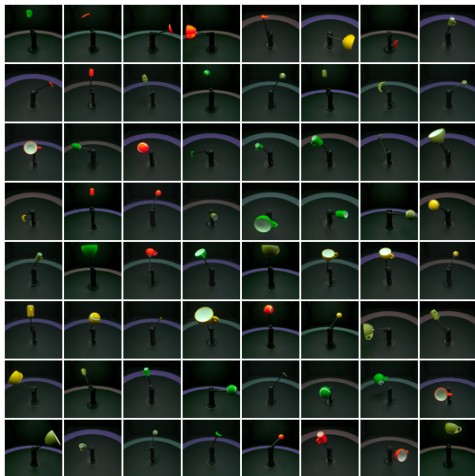

Figure 11: MPI 3D ($64 \times 64$)

c) Conversely, if for $z, \tilde{z} \in \mathbb{R}$ the $N(\rho, \sigma^2)$ measure of the line segment $[z, \tilde{z}]$ is preserved under the shift, i.e. $F(\tilde{z}_{\text{shifted}}) - F(z_{\text{shifted}}) = F(\tilde{z}) - F(z)$, then setting $z = -\infty$, we get $F(\tilde{z}_{\text{shifted}}) = F(\tilde{z}) + C$. $\qquad \square$

Notice also that $F(z_{\text{shifted}}) - F(z) = C$, so the three orange curvilinear rectangles on Figure 4 have the same area $C = 1/8$.

Recall that $F(z \mid \rho, \sigma^2) = \frac{1}{\sigma\sqrt{2\pi}} \int_{-\infty}^{z} \exp\left(-\frac{(t-\rho)^2}{2\sigma^2}\right) dt$ denotes here the cumulative function of the Gaussian distribution $N(\rho, \sigma^2)$.

## C  PROOF OF PROPOSITION 4.2

a) The shift defined by (1) for the distribution $q(z_i)$ acting on the latent space, preserves also any $q(z_j)$ for $j \neq i$. b) The result follows from the case of an arbitrary distribution over a pair of random variables $z_1, z_2$. For two variables, it follows from the Bayes formula that the shifts of $z_1$ preserve the conditional $q(z_2|z_1)$. Since the group(oid) action is transitive it follows that the conditional does not depend on $z_1$, and hence $q(z_1, z_2) = q(z_1)q(z_2)$.

## D  ARCHITECTURE DETAILS

In Table 2 we demonstrate VAE architecture. The Discriminator's architecture is described in Table 3.

For experiments with VAE+TopDis-C (Table 11), we used the following architecture configurations:

- dSprites: num channels $= 1, m_1 = 1, m_2 = 1, m_3 = 1, m_4 = 1, n = 5$;
- 3D Shapes: num channels $= 3, m_1 = 1, m_2 = 1, m_3 = 1, m_4 = 2, n = 5$.

For experiments with $\beta$-VAE+TopDis, $\beta$-TCVAE+TopDis, ControlVAE+TopDis, Factor-VAE+TopDis, DAVA+TopDis (Tables 1, 4), we used the following architecture configurations:

- dSprites: num channels $= 1, m_1 = 2, m_2 = 2, m_3 = 4, m_4 = 4, n = 5$;
- 3D Shapes: num channels $= 3, m_1 = 1, m_2 = 1, m_3 = 1, m_4 = 2, n = 5$;
- 3D Faces: num channels $= 1, m_1 = 1, m_2 = 1, m_3 = 1, m_4 = 2, n = 5$;
- MPI 3D: num channels $= 3, m_1 = 1, m_2 = 1, m_3 = 1, m_4 = 2, n = 6$.
- CelebA: num channels $= 3, m_1 = 1, m_2 = 1, m_3 = 1, m_4 = 2, n = 5$;

Table 2: Encoder and Decoder architecture for the dSprites experiments.

| Encoder | Decoder |
|---|---|
| Input: $64 \times 64 \times$ num channels | Input: $\mathbb{R}^{10}$ |
| $4 \times 4$ conv, 32 ReLU, stride 2 | $1 \times 1$ conv, $128 \times m_4$ ReLU, stride 1 |
| $4 \times 4$ conv, $32 \cdot m_1$ ReLU, stride 2 | $4 \times 4$ upconv, $64 \cdot m_3$ ReLU, stride 1 |
| $4 \times 4$ conv, $64 \cdot m_2$ ReLU, stride 2 | $4 \times 4$ upconv, $64 \cdot m_2$ ReLU, stride 2 |
| $4 \times 4$ conv, $64 \cdot m_3$ ReLU, stride 2 | $4 \times 4$ upconv, $32 \cdot m_1$ ReLU, stride 2 |
| $4 \times 4$ conv, $128 \cdot m_4$ ReLU, stride 1 | $4 \times 4$ upconv, 32 ReLU, stride 2 |
| $1 \times 1$ conv, $2 \times 10$, stride 1 | $4 \times 4$ upconv, 1, stride 2 |

Table 3: FactorVAE Discriminator architecture.

| Discriminator |
|---|
| [ FC, 1000 leaky ReLU ] $\times n$ |
| FC, 2 |

## E    RECONSTRUCTION ERROR

See Table 4.

## F    TRAINING CURVES

Figure 12 shows that TopDis loss decreases during training and has good negative correlation with MIG score, as expected. TopDis score was averaged in a sliding window of size 500, MIG was calculated every 50000 iterations.

## G    MORE ON RELATED WORK

Recently, approaches for learning disentangled representations through Hausdorff Factorized Support criterion (Roth et al., 2022) and adversarial learning (DAVA) (Estermann & Wattenhofer, 2023) were proposed.

Table 4: Reconstruction error.

| Method | dSprites | 3D Shapes | 3D Faces | MPI 3D |
|---|---|---|---|---|
| VAE | $8.67 \pm 0.29$ | $3494.10 \pm 3.27$ | $1374.42 \pm 3.38$ | $3879.11 \pm 0.76$ |
| $\beta$-TCVAE | $17.87 \pm 0.56$ | $3492.25 \pm 5.79$ | $1375.03 \pm 3.41$ | $3891.03 \pm 1.41$ |
| $\beta$-TCVAE + TopDis (ours) | $17.32 \pm 0.31$ | $3495.13 \pm 2.49$ | $1376.21 \pm 3.09$ | $3889.34 \pm 1.97$ |
| $\beta$-VAE | $12.97 \pm 0.50$ | $3500.60 \pm 13.59$ | $1379.64 \pm 0.19$ | $3888.75 \pm 2.27$ |
| $\beta$-VAE + TopDis (ours) | $13.75 \pm 0.63$ | $3495.76 \pm 6.54$ | $1380.10 \pm 0.19$ | $3886.61 \pm 0.97$ |
| ControlVAE | $15.32 \pm 0.47$ | $3499.61 \pm 12.13$ | $1404.42 \pm 5.01$ | $3889.81 \pm 0.43$ |
| ControlVAE + TopDis (ours) | $14.91 \pm 0.39$ | $3500.28 \pm 10.73$ | $1389.42 \pm 4.47$ | $3889.24 \pm 0.50$ |
| FactorVAE | $14.65 \pm 0.41$ | $3501.53 \pm 13.43$ | $1488.26 \pm 4.47$ | $3884.57 \pm 0.52$ |
| FactorVAE + TopDis (ours) | $14.72 \pm 0.49$ | $3504.42 \pm 9.98$ | $1377.93 \pm 3.47$ | $3886.10 \pm 0.36$ |
| DAVA | $36.41 \pm 2.03$ | $3532.56 \pm 14.14$ | $1403.77 \pm 0.99$ | $3902.54 \pm 1.41$ |
| DAVA + TopDis (ours) | $26.03 \pm 2.51$ | $3537.39 \pm 40.52$ | $1403.20 \pm 0.49$ | $3898.54 \pm 1.33$ |

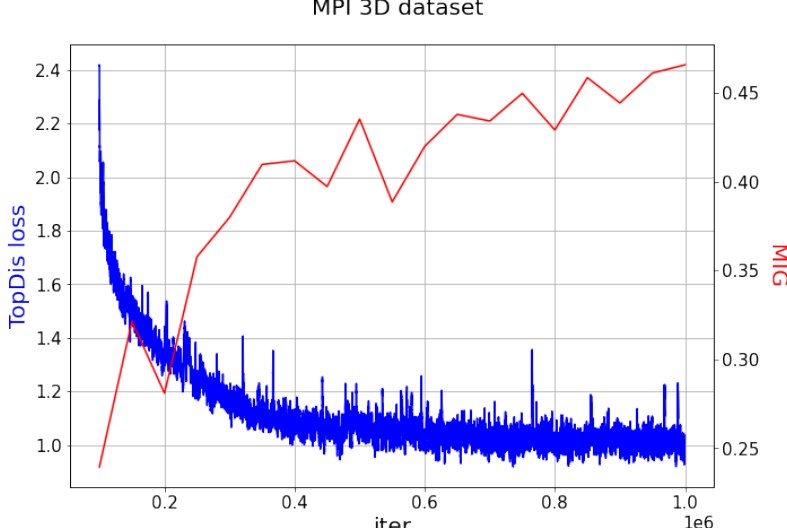

Figure 12: MPI 3D: Training curves of TopDis loss and MIG.

Table 5: Comparison with a recent state-of-the-art method

| Method | FactorVAE score | MIG | SAP | DCI, dis. |
|---|---|---|---|---|
| | dSprites | | | |
| TCWAE | $0.76 \pm 0.03$ | $0.32 \pm 0.04$ | $0.072 \pm 0.004$ | - |
| FactorVAE + TopDis (ours) | $\mathbf{0.82 \pm 0.04}$ | $\mathbf{0.36 \pm 0.03}$ | $\mathbf{0.082 \pm 0.001}$ | $\mathbf{0.52 \pm 0.04}$ |

The paper Barannikov et al. (2022) describes briefly an application of topological metric to evaluation of interpretable directions in a simple synthetic dataset. They compare topological dissimilarity in data submanifolds corresponding to slices in the latent space, while we use axis-aligned traversals and samples from the whole data manifold. More importantly, we develop a differentiable pipeline for VAE to learn disentangled representation from scratch. Also we use group action shifts and gradient orthogonalization.

In recent works Farajtabar et al. (2020); Suteu & Guo (2019), the authors propose to use a technique of gradient orthogonalization to overcome the problem of multi-task optimization. The main idea behind gradient orthogonalization is to modify the gradients of different tasks in a way that they become more orthogonal to each other, thus reducing conflicts during the optimization process.

In Table 5, we compare our results with another recent state-of-the-art method, TCWAE (Gaujac et al., 2021). Since there is no code available to replicate their results, we present the values from the original papers. The architecture and training setup were essentially identical to what is described in this paper.

## H   VAE + TOPDIS-C

We also explore TopDis as self-sufficient loss. Specifically, we add simply our TopDis loss to the classical VAE objective. We found that it is beneficial in this setting to use TopDis in contrastive learning setup, when we not only minimize the topological dissimilarity between shifted point clouds, but also maximize the difference in topology structures if the shift is made not along one latent code for all points in a batch but in random directions in latent space. We call this variant of our method TopDis-C.

In Table 11 we demonstrate that the TopDis-C loss significantly improves, even without total correlation loss, the disentanglement quality of simple VAE. See details of architecture and training

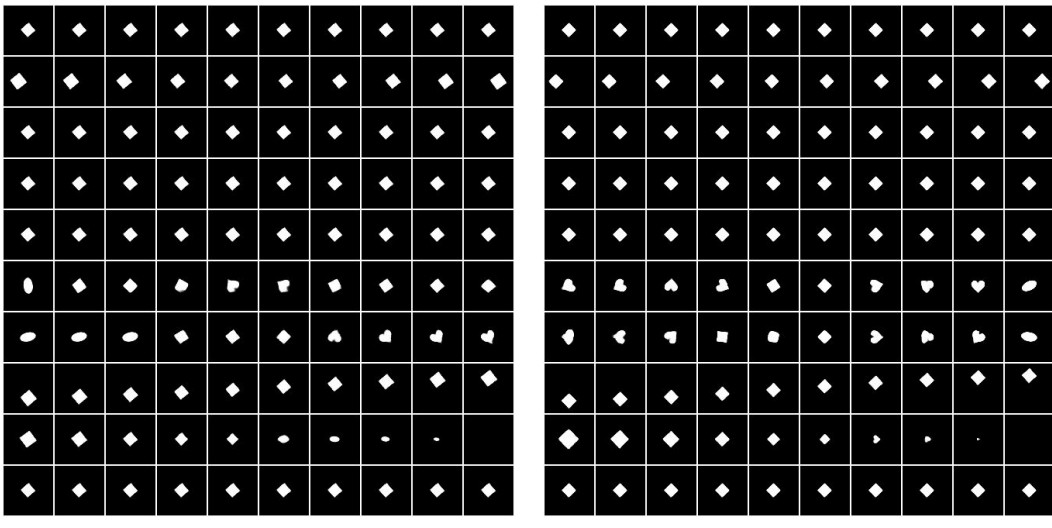

(a) VAE, dSprites

(b) VAE + TopDis-C, dSprites

Figure 13: VAE and VAE + TopDis-C latent traversals, dSprites.

Table 6: Evaluation of the proposed VAE + TopDis-C on the benchmark datasets.

| Method | MIG | DCI, dis. | Reconstruction error |
|---|---|---|---|
| dSprites | | | |
| VAE | $0.161 \pm 0.058$ | $0.301 \pm 0.070$ | $\mathbf{21.24 \pm 3.34}$ |
| VAE + TopDis-C (ours) | $\mathbf{0.242 \pm 0.019}$ | $\mathbf{0.430 \pm 0.027}$ | $22.67 \pm 4.87$ |
| 3D Shapes | | | |
| VAE | $0.729 \pm 0.070$ | $0.952 \pm 0.023$ | $3494.10 \pm 3.27$ |
| VAE + TopDis-C (ours) | $\mathbf{0.773 \pm 0.048}$ | $\mathbf{0.990 \pm 0.006}$ | $\mathbf{3484.96 \pm 3.40}$ |

in Sections D, L. We plot all latent traversals in Figures 13a, 13b for dSprites dataset, that confirm quantitative results with visual perception, see e.g. the entangling of the size increase along the x- and y-shifts traversals in raws 2, 8 on Figure 13a.

## I  UNSUPERVISED DISCOVERY OF DISENTANGLED DIRECTIONS IN STYLEGAN

We perform additional experiments to study the ability of the proposed topology-based loss to infer disentangled directions in a pretrained GAN. In experiments, we used StyleGAN (Karras et al., 2019)[3]. The unsupervised directions were explored in the style space $\mathcal{Z}$. To filter out non-informative directions we followed the approach from Härkönen et al. (2020) and selected top 32 directions by doing PCA for the large batch of data in the style space. Then, we selected the new basis $n_i$, $i = 1, \ldots, 32$ in this subspace, starting from a random initialization. Directions $n_i$ were selected sequentially by minimization of RTD along shifts in $\mathcal{Z}$ space:

$$\mathrm{RTD}(Gen_k(Z), Gen_k(Z + cn_i)),$$

where $Gen_k(\cdot)$ is the $k-$layer of the StyleGAN generator (we used $k = 3$). After each iteration the Gram–Schmidt orthonormalization process for $n_i$ was performed. We were able to discover at least 3 disentangled directions: azimuth (Fig. 14), smile (Fig. 15), hair color (Fig. 16).

---

[3]we used a PyTorch reimplementation from:
https://github.com/rosinality/style-based-gan-pytorch.

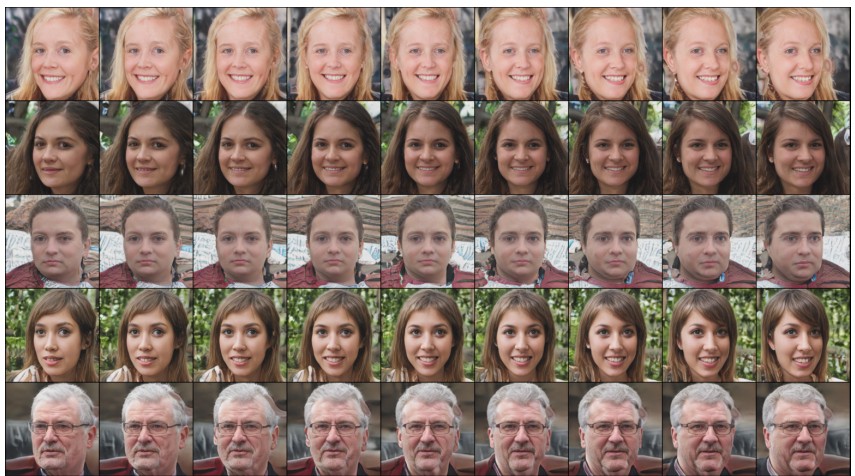

Figure 14: StyleGAN, change of an azimuth.

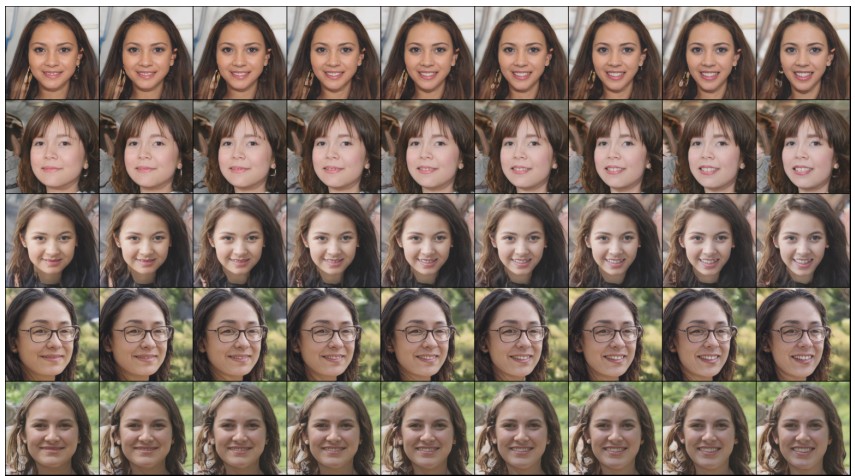

Figure 15: StyleGAN, change of a smile.

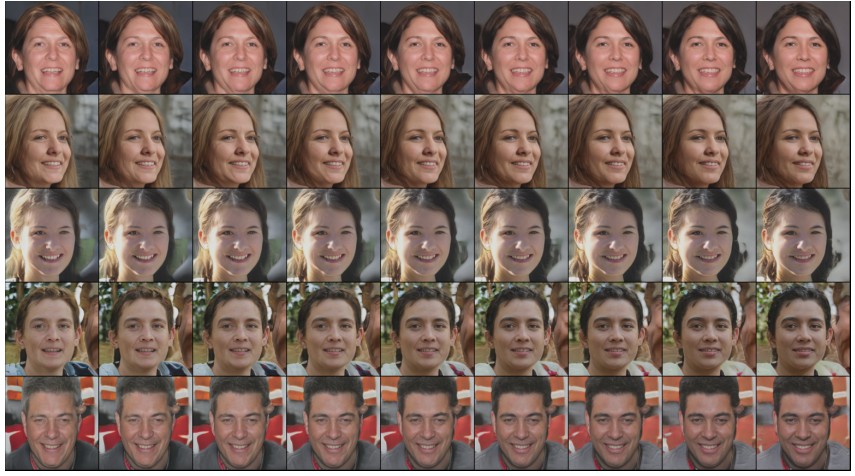

Figure 16: StyleGAN, change of a hair color.

Table 7: Evaluation of the proposed FactorVAE + TopDis on the benchmark datasets for separate runs.

| Method | FactorVAE score | | | MIG | | | SAP | | | DCI, dis. | | |
|---|---|---|---|---|---|---|---|---|---|---|---|---|
| | run 1 | run 2 | run 3 | run 1 | run 2 | run 3 | run 1 | run 2 | run 3 | run 1 | run 2 | run 3 |
| dSprites | | | | | | | | | | | | |
| FactorVAE | **0.856** | 0.830 | 0.786 | **0.341** | 0.308 | 0.243 | 0.054 | 0.053 | 0.051 | **0.565** | 0.526 | **0.509** |
| FactorVAE + TopDis (ours) | 0.779 | **0.845** | **0.847** | 0.331 | **0.382** | **0.360** | **0.082** | **0.081** | **0.092** | 0.489 | **0.571** | 0.503 |
| 3D Shapes | | | | | | | | | | | | |
| FactorVAE | 0.901 | 0.893 | **1.000** | 0.678 | 0.573 | **0.867** | 0.055 | 0.067 | 0.175 | 0.780 | 0.772 | **0.996** |
| FactorVAE + TopDis (ours) | **0.924** | **1.000** | **1.000** | **0.810** | **0.787** | 0.739 | **0.123** | **0.172** | **0.184** | **0.837** | **0.991** | 0.991 |
| 3D Faces | | | | | | | | | | | | |
| FactorVAE | **1.000** | **1.000** | **1.000** | 0.597 | 0.533 | 0.649 | **0.059** | 0.048 | 0.076 | 0.843 | 0.840 | 0.861 |
| FactorVAE + TopDis (ours) | **1.000** | **1.000** | **1.000** | **0.631** | **0.596** | **0.651** | 0.058 | **0.051** | **0.077** | **0.859** | 0.835 | **0.907** |
| MPI 3D | | | | | | | | | | | | |
| FactorVAE | 0.582 | 0.651 | 0.650 | 0.323 | 0.388 | 0.341 | 0.160 | 0.230 | **0.239** | 0.379 | 0.435 | 0.430 |
| FactorVAE + TopDis (ours) | **0.696** | **0.662** | **0.674** | **0.455** | **0.416** | **0.389** | **0.283** | **0.259** | 0.221 | **0.505** | **0.464** | **0.437** |

## J  DATASETS

The dSprites dataset is a collection of 2D shapes generated procedurally from five independent latent factors: shape, scale, rotation, x-coordinate, and y-coordinate of a sprite. The 3D Shapes dataset, on the other hand, consists of 3D scenes with six generative factors: floor hue, wall hue, orientation, shape, scale, and shape color. The 3D Faces dataset consists of 3D rendered faces with four generative factors: face id, azimuth, elevation, lighting. The MPI 3D dataset contains images of physical 3D objects with seven generative factors: color, shape, size, camera height, background color, horizontal and vertical axes. We used the MPI3D-Real dataset with images of complex shapes from the robotic platform provides a trade-off between real-world data with unknown underlying factors of variations and a more controlled dataset allowing quantitative evaluation.

These datasets were chosen as they provide a diverse range of images and have well-defined disentangled factors, making them suitable for evaluating the performance of our proposed method.

We also evaluate our method on CelebA dataset that provides images of aligned faces of celebrities. This dataset doesn't have any ground truth generative factors because of its real-world nature.

## K  MORE DETAILS ON THE SIGNIFICANCE OF TOPDIS EFFECT

In order to accurately assess the impact of the TopDis term, we employed a consistent set of random initializations. This approach was adopted to eliminate potential confounding factors that may arise from disparate initial conditions. This allowed us to attribute any observed improvements in disentanglement quality specifically to the inclusion of the TopDis term in our model. In Table 7 we demonstrate the consistent improvement across multiple runs.

## L  TRAINING DETAILS

Following the previous work Kim & Mnih (2018), we used similar architectures for the encoder, decoder and discriminator, the same for all models. We set the latent space dimensionality to 10.

We normalized the data to $[0, 1]$ interval and trained 1M iterations with batch size of $64$ and Adam (Kingma & Ba, 2015) optimizer. The learning rate for VAE updates was $10^{-4}$ for dSprites and MPI 3D datasets, $10^{-3}$ for 3D Shapes dataset, and $2 \times 10^{-4}$ for 3D faces and CelebA datasets, $\beta_1 = 0.9$, $\beta_2 = 0.999$, while the learning rate for discriminator updates was $10^{-4}$ for dSprites, 3D Faces, MPI 3D and CelebA datasets, $10^{-3}$ for 3D Shapes dataset, $\beta_1 = 0.5$, $\beta_2 = 0.9$ for discriminator updates. In order to speed up convergence on MPI 3D, we first trained the model with FactorVAE loss only for 100000 iterations and then continued training with TopDis loss. We also fine-tuned the hyperparameter $\gamma$ over set commonly used in the literature (Kim & Mnih, 2018; Locatello et al., 2019; Ridgeway & Mozer, 2018) to achieve the best performance on the baseline models.

The best performance found hyperparameters are the following:

- dSprites. $\beta$-TCVAE: $\beta = 6$, $\beta$-TCVAE+ TopDis: $\beta = 6, \gamma = 5$ $\beta$-VAE: $\beta = 2$, $\beta$-VAE + TopDis: $\beta = 2, \gamma = 4$, FactorVAE: $\gamma = 20$, FactorVAE + TopDis: $\gamma_1 = 5, \gamma_2 = 5$, DAVA + TopDis: $\gamma = 5$;

- 3D Shapes. $\beta$-TCVAE: $\beta = 4$, $\beta$-TCVAE + TopDis: $\beta = 4, \gamma = 5$, $\beta$-VAE: $\beta = 2$, $\beta$-VAE + TopDis: $\beta = 2, \gamma = 1$, FactorVAE: $\gamma = 30$, FactorVAE + TopDis: $\gamma_1 = 5, \gamma_2 = 5$, DAVA + TopDis: $\gamma = 3$;

- 3D Faces. $\beta$-TCVAE: $\beta = 6$, $\beta$-TCVAE + TopDis: $\beta = 6, \gamma = 5$, $\beta$-VAE: $\beta = 2$, $\beta$-VAE + TopDis: $\beta = 2, \gamma = 1$, FactorVAE: $\gamma = 5$, FactorVAE + TopDis: $\gamma_1 = 5, \gamma_2 = 5$, DAVA + TopDis: $\gamma = 2$;

- MPI 3D. $\beta$-TCVAE: $\beta = 6$, $\beta$-TCVAE + TopDis: $\beta = 6, \gamma = 5$, $\beta$-VAE: $\beta = 2$, $\beta$-VAE + TopDis: $\beta = 2, \gamma = 1$, FactorVAE: $\gamma = 10$, FactorVAE + TopDis: $\gamma_1 = 5, \gamma_2 = 6$, DAVA + TopDis: $\gamma = 5$;

- CelebA. FactorVAE: $\gamma = 5$, FactorVAE + TopDis: $\gamma_1 = 5, \gamma_2 = 2$;

For the ControlVAE and ControlVAE+TopDis experiments[4], we utilized the same set of relevant hyperparameters as in the FactorVAE and FactorVAE+TopDis experiments. Additionally, ControlVAE requires an expected KL loss value as a hyperparameter, which was set to KL=18, as in the original paper. It should also be noted that the requirement of an expected KL loss value is counterintuitive for an unsupervised problem, as this value depends on the number of true factors of variation. For the DAVA and DAVA + TopDis experiments[5], we used the original training procedure proposed in (Estermann & Wattenhofer, 2023), adjusting the batch size to $64$ and number of iteration to $1000000$ to match our setup .

## M    Computational complexity

The complexity of the $\mathcal{L}_{TD}$ is formed by the calculation of RTD. For the batch size $N$, object dimensionality $C \times H \times W$ and latent dimensionality $d$, the complexity is $O(N^2(CHW + d))$, because all the pairwise distances in a batch should be calculated. The calculation of the RTD itself is often quite fast for batch sizes $\leq 256$ since the boundary matrix is typically sparse for real datasets (Barannikov et al., 2022). Operation, required to RTD differentiation do not take extra time. For RTD calculation and differentiation, we used GPU-optimized software.

## N    Formal definition of Representation Topology Divergence (RTD)

Data points in a high-dimensional space are often concentrated near a low-dimensional manifold (Goodfellow et al., 2016). The manifold's topological features can be represented via Vietoris-Rips simplicial complex, a union of simplices whose vertices are points at a distance smaller than a threshold $\alpha$.

---

[4]`https://github.com/shj1987/ControlVAE-ICML2020`.
[5]`https://github.com/besterma/dava`

We define the weighted graph $\mathcal{G}$ with data points as vertices and the distances between data points $d(A_i A_j)$ as edge weights. The Vietoris-Rips complex at the threshold $\alpha$ is then:

$$\text{VR}_\alpha(\mathcal{G}) = \{\{A_{i_0}, \ldots, A_{i_k}\}, A_i \in \text{Vert}(\mathcal{G}) \mid d(A_i A_j) \leq \alpha\},$$

The vector space $C_k$ consists of all formal linear combinations of the $k$-dimensional simplices from $\text{VR}_\alpha(\mathcal{G})$ with modulo 2 arithmetic. The boundary operators $\partial_k : C_k \to C_{k-1}$ maps each simplex to the sum of its facets. The $k$-th homology group $H_k = ker(\partial_k)/im(\partial_{k+1})$ represents $k-$dimensional topological features.

Choosing $\alpha$ is challenging, so we analyze all $\alpha > 0$. This creates a filtration of nested Vietoris-Rips complexes. We track the "birth" and "death" scales, $\alpha_b, \alpha_d$, of each topological feature, defining its persistence as $\alpha_d - \alpha_b$. The sequence of the intervals $[\alpha_b, \alpha_d]$ for basic features forms the persistence barcode (Barannikov, 1994; Chazal & Michel, 2017).

The standard persistence barcode analyzes a single point cloud $X$. The Representation Topology Divergence (RTD) (Barannikov et al., 2022) was introduced to measure the multi-scale topological dissimilarity between two point clouds $X, \tilde{X}$. This is done by constructing an auxilary graph $\hat{\mathcal{G}}^{w,\tilde{w}}$ whose Vietoris-Rips complex measures the difference between Vietoris-Rips complexes $\text{VR}_\alpha(\mathcal{G}^w)$ and $\text{VR}_\alpha(\mathcal{G}^{\tilde{w}})$, where $w, \tilde{w}$ are the distance matrices of $X, \tilde{X}$. The auxiliary graph $\hat{\mathcal{G}}^{w,\tilde{w}}$ has the double set of vertices and the edge weights matrix $\begin{pmatrix} 0 & (w_+)^\intercal \\ w_+ & \min(w, \tilde{w}) \end{pmatrix}$, where $w_+$ is the $w$ matrix with lower-triangular part replaced by $+\infty$.

The *R-Cross-Barcode$_k(X, \tilde{X})$* is the persistence barcode of the filtered simplicial complex $\text{VR}(\hat{\mathcal{G}}^{w,\tilde{w}})$. $\text{RTD}_k(X, \tilde{X})$ equals the sum of intervals' lengths in *R-Cross-Barcode$_k(X, \tilde{X})$* and measures its closeness to an empty set, with longer lifespans indicating essential features. $\text{RTD}(X, \tilde{X})$ is the half-sum $\text{RTD}(X, \tilde{X}) = \frac{1}{2}(\text{RTD}_1(X, \tilde{X}) + \text{RTD}_1(\tilde{X}, X))$.

## O  SYMMETRY GROUP(OID) ACTION

A groupoid is a mathematical structure that generalizes the concept of a group. It consists of a set $G$ along with a partially defined binary operation. Unlike groups, the binary operation in a groupoid is not required to be defined for all pairs of elements. More formally, a groupoid is a set $G$ together with a binary operation $\cdot : G \times G \to G$ that satisfies the following conditions for all $a, b, c$ in $G$ where the operations are defined: 1) Associativity: $(a \cdot b) \cdot c = a \cdot (b \cdot c)$; 2) Identity: there is an element $e$ in $G$ such that $a \cdot e = e \cdot a = a$ for each $a$ in $G$; 3) Inverses: for each $a$ in $G$, there is an element $a^{-1}$ in $G$ such that $a \cdot a^{-1} = a^{-1} \cdot a = e$.

A Lie groupoid is a groupoid that has additional structure of a manifold, together with smooth structure maps. These maps are required to satisfy certain properties analogous to those of a groupoid, but in a smooth category. See (Weinstein (1996)) for details.

## P  ABLATION STUDY

We have performed the experiments concerning the ablation study of gradient orthogonalization technique. First, we evaluate the effect of gradient orthogonalization when integrating TopDis into the classical VAE model on dSprites, see Figure P and Table 8. We conduct this experiment to verify the gradient orthogonalization technique in the basic setup when additional terms promoting disentanglement are absent. Second, we evaluate the effect of gradient orthogonalization when integrating TopDis to FactorVAE on the MPI3D dataset. This experiment verifies how gradient orthogonalization works for more complex data in the case of a more complicated objective. We highlight that adding the gradient orthogonalization results in lower reconstruction loss throughout the training. In particular, this may be relevant when the reconstruction quality is of high importance.

## Q  SENSITIVITY ANALYSIS

We provide the sensitivity analysis w.r.t. $\gamma$ from equation 4 for FactorVAE+TopDis on MPI3D-Real ($3 \cdot 10^5$ training iterations), please see Table 9. In Table 9, $\gamma_{TD}$ denotes the weight $\gamma$ for the TopDis

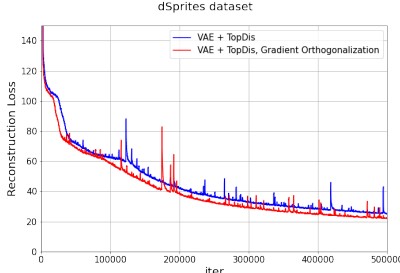 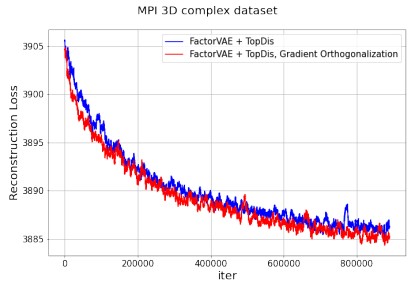

Figure 17: Effect of gradient orthogonalization on reconstruction loss. Left: VAE+TopDis, dSprites. Right: FactorVAE+TopDis, MPI 3D.

Table 8: Effect of gradient orthogonalization on disentanglement.

| Method | FactorVAE | MIG | SAP | DCI, dis. |
|---|---|---|---|---|
| dSprites | | | | |
| VAE + TopDis, no gradient orthogonalization | 0.736 | 0.098 | 0.041 | 0.202 |
| VAE + TopDis, gradient orthogonalization | 0.723 | 0.121 | 0.031 | 0.229 |
| MPI 3D | | | | |
| FactorVAE + TopDis, no gradient orthogonalization | 0.696 | 0.455 | 0.283 | 0.505 |
| FactorVAE + TopDis, gradient orthogonalization | 0.707 | 0.466 | 0.288 | 0.508 |

loss from equation 4 while $\gamma_{TC}$ denotes the weight for the Total Correlation loss from the FactorVAE model (see Kim & Mnih (2018) for details). In particular, $\gamma_{TC} = 5, \gamma_{TD} = 0$ corresponds to plain FactorVAE model.

Table 9: Sensitivity analysis. $\gamma_{TD}$ denotes the weight for the TopDis loss (see equation 4 for details) while $\gamma_{TC}$ denotes the weight for the Total Correlation loss from the FactorVAE model (see Kim & Mnih (2018) for details).

| Method | FactorVAE | MIG | SAP | DCI, dis. | Reconstruction |
|---|---|---|---|---|---|
| FactorVAE + TopDis, MPI 3D | | | | | |
| $\gamma_{TC} = 5, \gamma_{TD} = 0$ | $0.586 \pm 0.038$ | $0.300 \pm 0.020$ | $0.184 \pm 0.028$ | $0.357 \pm 0.001$ | $3888.96 \pm 0.94$ |
| $\gamma_{TC} = 5, \gamma_{TD} = 3$ | $0.607 \pm 0.047$ | $0.320 \pm 0.003$ | $0.193 \pm 0.015$ | $0.401 \pm 0.025$ | $3891.28 \pm 0.91$ |
| $\gamma_{TC} = 5, \gamma_{TD} = 5$ | $0.605 \pm 0.048$ | $0.332 \pm 0.033$ | $0.205 \pm 0.025$ | $0.397 \pm 0.038$ | $3892.51 \pm 1.19$ |
| $\gamma_{TC} = 5, \gamma_{TD} = 6$ | $0.605 \pm 0.051$ | $0.340 \pm 0.035$ | $0.207 \pm 0.036$ | $0.412 \pm 0.026$ | $3892.17 \pm 0.54$ |
| $\gamma_{TC} = 5, \gamma_{TD} = 7$ | $0.594 \pm 0.041$ | $0.297 \pm 0.042$ | $0.183 \pm 0.029$ | $0.362 \pm 0.050$ | $3892.98 \pm 0.70$ |

## R  RTD DIFFERENTIATION

Here we gather details on RTD differentiation in order to use RTD as a loss in neural networks.

Define $\Sigma$ as the set of all simplices in the filtration of the graph $VR(\hat{\mathcal{G}}^{w,\tilde{w}})$, and $\mathcal{T}_k$ as the set of all segments in *R-Cross-Barcode$_k$(X, $\hat{X}$)*. Fix (an arbitrary) strict order on $\mathcal{T}_k$.

There exists a function $f_k : \{b_i, d_i\}_{(b_i, d_i) \in \mathcal{T}_k} \to \Sigma$ that maps $b_i$ (or $d_i$) to simplices $\sigma$ (or $\tau$) whose addition leads to "birth" (or "death") of the corresponding homological class.

Thus, we may obtain the following equation for subgradient

$$\frac{\partial \operatorname{RTD}(X, \hat{X})}{\partial \sigma} = \sum_{i \in \mathcal{T}_k} \frac{\partial \operatorname{RTD}(X, \hat{X})}{\partial b_i} \mathbb{I}\{f_k(b_i) = \sigma\} + \sum_{i \in \mathcal{T}_k} \frac{\partial \operatorname{RTD}(X, \hat{X})}{\partial d_i} \mathbb{I}\{f_k(d_i) = \sigma\}$$

Here, for any $\sigma$ no more than one term has non-zero indicator.

$b_i$ and $d_i$ are just the filtration values at which simplices $f_k(b_i)$ and $f_k(d_i)$ join the filtration. They depend on weights of graph edges as

$$g_k(\sigma) = \max_{i,j \in \sigma} m_{i,j}$$

This function is differentiable (Leygonie et al., 2021) and so is $f_k \circ g_k$. Thus we obtain the subgradient:

$$\frac{\partial \operatorname{RTD}(X, \hat{X})}{\partial m_{i,j}} = \sum_{\sigma \in \Sigma} \frac{\partial \operatorname{RTD}(X, \hat{X})}{\partial \sigma} \frac{\partial \sigma}{\partial m_{i,j}}.$$

The only thing that is left is to obtain subgradients of $\operatorname{RTD}(X, \hat{X})$ by points from $X$ and $\hat{X}$. Consider (an arbitrary) element $m_{i,j}$ of matrix $m$. There are 4 possible scenarios:

1. $i, j \leq N$, in other words $m_{i,j}$ is from the upper-left quadrant of $m$. Its length is constant and thus $\forall l : \frac{\partial m_{i,j}}{\partial X_l} = \frac{\partial m_{i,j}}{\partial \hat{X}_l} = 0$.

2. $i \leq N < j$, in other words $m_{i,j}$ is from the upper-right quadrant of $m$. Its length is computed as Euclidean distance and thus $\frac{\partial m_{i,j}}{\partial X_i} = \frac{X_i - \hat{X}_{j-N}}{\|X_i - \hat{X}_{j-N}\|_2}$ (similar for $X_{N-j}$).

3. $j \leq N < i$, similar to the previous case.

4. $N < i, j$, in other words $m_{i,j}$ is from the bottom-right quadrant of $m$. Here we have subgradients like

$$\frac{\partial m_{i,j}}{\partial X_{i-N}} = \frac{X_{i-N} - X_{j-N}}{\|X_{i-N} - X_{j-N}\|_2} \mathbb{I}\{w_{i-N,j-N} < \hat{w}_{i-N,j-N}\}$$

Similar for $X_{j-N}, \hat{X}_{i-N}$ and $\hat{X}_{j-N}$.

Subgradients $\frac{\partial \operatorname{RTD}(X, \hat{X})}{\partial X_i}$ and $\frac{\partial \operatorname{RTD}(X, \hat{X})}{\partial \hat{X}_i}$ can be derived from the before mentioned using the chain rule and the formula of full (sub)gradient. Now we are able to minimize $\operatorname{RTD}(X, \hat{X})$ by methods of (sub)gradient optimization.

## S   DISCUSSING THE DEFINITION OF DISENTANGLED REPRESENTATION.

Let $X \subset \mathbb{R}^{N_x \times N_y}$ denotes the dataset consisting of $N_x \times N_y$ pixels pictures containing a disk of various color with fixed disk radius $r$ and the center of the disks situated at an arbitrary point $x, y$. Denote $\rho_X$ the uniform distribution over the coordinates of centers of the disks and the colors. Let $G_x \times G_y \times G_c$ be the commutative group of symmetries of this data distribution, $G_x \times G_y$ is the position change acting (locally) via

$$(a, b) : (x, y, c) \mapsto (x + a, y + b, c)$$

and $G_z$ is changing the colour along the colour circle $\theta : (x, y, c) \mapsto (x, y, c + \theta \mod 2\pi)$. Contrary to Higgins et al. (2018), section 3, we do not assume the gluing of the opposite sides of our pictures, which is closer to real world situations. Notice that, as a consequence of this, each group element from $G_x \times G_y$ can act only on a subset of $X$, so that the result is still situated inside $N_x \times N_y$ pixels picture. This mathematical structure when each group element has its own set of points on which it acts, is called groupoid, we discuss this notion in more details in Appendix O.

The outcome of disentangled learning in such case are the encoder $h : X \to Z$ and the decoder $f : Z \to X$ maps with $Z = \mathbb{R}^3$, $f \circ h = \operatorname{Id}$, together with symmetry group(oid) $G$ actions on $X$

Table 10: Evaluation on the benchmark datasets with correlated factors

| Method | FactorVAE score | MIG | SAP | DCI, dis. |
|---|---|---|---|---|
| dSprites | | | | |
| FactorVAE | $0.803 \pm 0.055$ | $0.086 \pm 0.026$ | $0.030 \pm 0.010$ | $0.216 \pm 0.044$ |
| FactorVAE + TopDis (ours) | $\mathbf{0.840 \pm 0.011}$ | $\mathbf{0.103 \pm 0.019}$ | $\mathbf{0.044 \pm 0.014}$ | $\mathbf{0.270 \pm 0.002}$ |
| 3D Shapes | | | | |
| FactorVAE | $0.949 \pm 0.67$ | $0.363 \pm 0.100$ | $0.083 \pm 0.004$ | $0.477 \pm 0.116$ |
| FactorVAE + TopDis (ours) | $\mathbf{0.998 \pm 0.001}$ | $\mathbf{0.403 \pm 0.091}$ | $\mathbf{0.112 \pm 0.013}$ | $\mathbf{0.623 \pm 0.026}$ |

and $Z$, such that a) the encoder-decoder maps preserve the distributions, which are the distribution $\rho_X$ describing the dataset $X$ and the standard in VAE learning $N(0,1)$ distribution in latent space $Z$; b) the decoder and the encoder maps are equivariant with respect to the symmetry group(oid) action, where the action on the latent space is defined as shifts of latent variables; the group action preserves the dataset distribution $X$ therefore the group(oid) action shifts on the latent space must preserve the standard $N(0,1)$ distribution on latent coordinates, i.e. they must act via the formula 1.

**Connection with disentangled representations in which the symmetry group latent space action is *linear*.** The normal distribution arises naturally as the projection to an axis of the uniform distribution on a very high dimensional sphere $S^N \subset \mathbb{R}^{N+1}$. Let a general symmetry compact Lie group $\hat{G}$ acts linearly on $\mathbb{R}^{N+1}$ and preserves the sphere $S^N$. Let $G^{ab}$ be a maximal commutative subgroup in $G$. Then the ambient space $\mathbb{R}^{N+1}$ decomposes into direct sum of subspaces $\mathbb{R}^{N+1} = \oplus_\alpha Z_\alpha$, on which $G^{ab} = \Pi_i G_i$, acts via rotations in two-dimensional space, and the orbit of this action is a circle $S^1 \subset S^N$. If one chooses an axis in each such two-dimensional space then the projection to this axis gives a coordinate on the sphere $S^N$. And the group action of $G^{ab}$ decomposes into independent actions along these axes. In such a way, the disentangled representation in the sense of Section 4.1 can be obtained from the data representation with uniform distribution on the sphere/disk on which the symmetry group action is linear, and vice versa.

## T  ON EQUIVALENCE OF SYMMETRY BASED AND FACTORS INDEPENDENCE BASED DEFINITIONS OF DISENTANGLEMENT

**Proposition T.1.**  *Assume that the variational autoencoder satisfies the conditions listed in Section 4.1. Then it satisfies the conditions of the "factors independence definition" and vice versa.*

## U  EXPERIMENTS WITH CORRELATED FACTORS

Table 10 shows experimental results for disentanglement learning with confounders - one factor correlated with all others. The addition of the TopDis loss results in a consistent improvement of all quality measures. For experiments, we used the implementation of the "shared confounders" distribution from Roth et al. (2022)[6] and the same hyperparameters as for the rest of experiments.

## V  VISUALIZATION OF LATENT TRAVERSALS

Images obtained from selected latent traversal exhibiting the most differences are presented in Figure 18 (FactorVAE, FactorVAE+TopDis trained on dSprites, 3D shapes, MPI 3D, 3D Faces) and Figure 19 (FactorVAE, FactorVAE+TopDis trained on CelebA).

Figures 20, 21, 22 shows latent traversals along all axes.

---

[6] https://github.com/facebookresearch/disentangling-correlated-factors

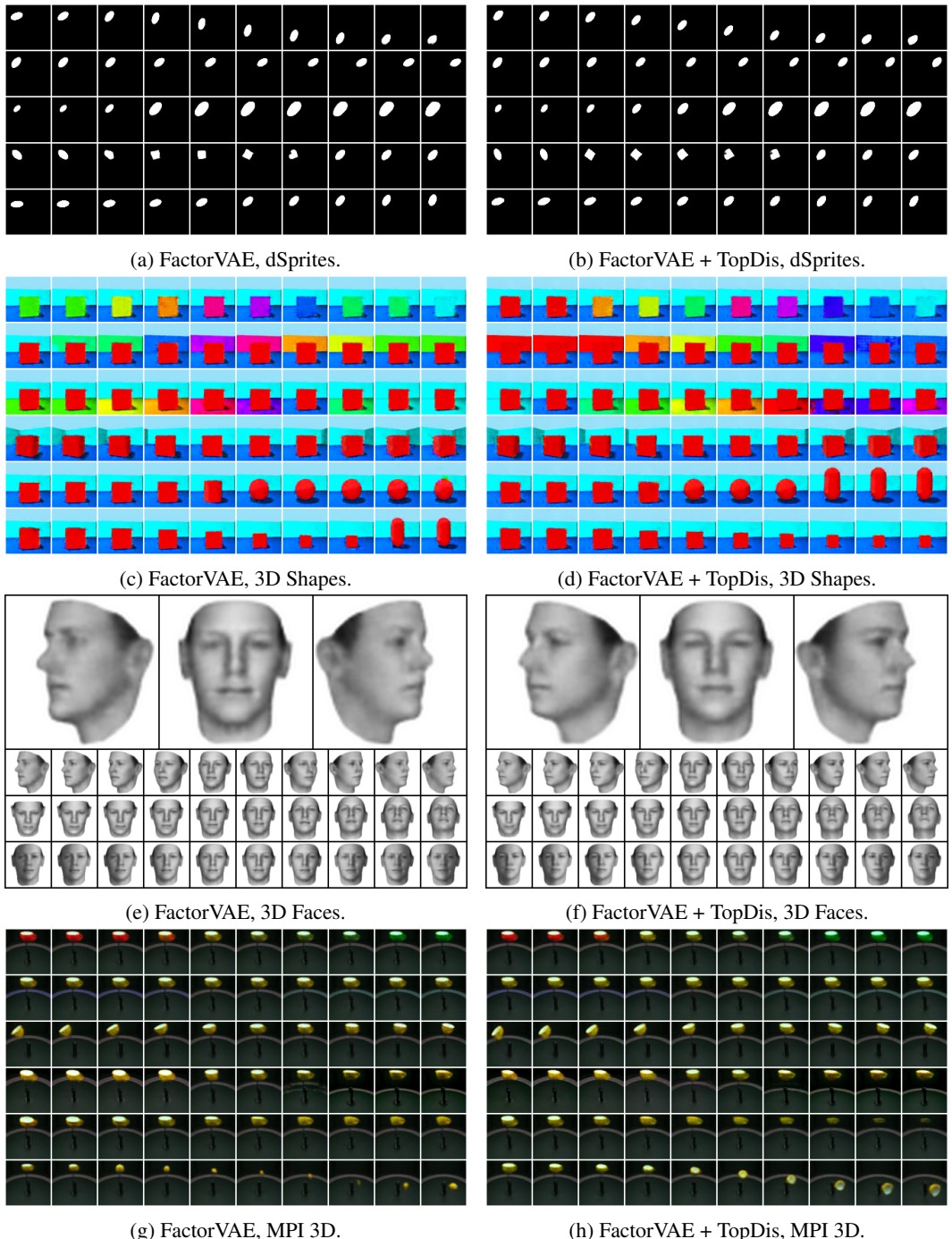

(a) FactorVAE, dSprites.

(b) FactorVAE + TopDis, dSprites.

(c) FactorVAE, 3D Shapes.

(d) FactorVAE + TopDis, 3D Shapes.

(e) FactorVAE, 3D Faces.

(f) FactorVAE + TopDis, 3D Faces.

(g) FactorVAE, MPI 3D.

(h) FactorVAE + TopDis, MPI 3D.

Figure 18: FactorVAE and FactorVAE + TopDis latent traversals.

# W    EXPERIMENTS WITH VAE + TOPDIS

In this section we explore TopDis as the only regularization added to the classical VAE objective. For results, see 11.

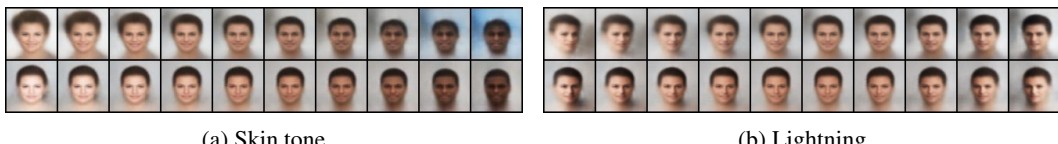

(a) Skin tone            (b) Lightning

Figure 19: Visual improvement from addition of TopDis, CelebA. Top: FactorVAE, bottom: Factor-VAE + TopDis.

Table 11: Evaluation on the benchmark datasets for VAE + TopDis

| Method | FactorVAE score | MIG | SAP | DCI, dis. |
|---|---|---|---|---|
| dSprites | | | | |
| VAE | $0.781 \pm 0.016$ | $0.170 \pm 0.072$ | $0.057 \pm 0.039$ | $0.314 \pm 0.072$ |
| VAE + TopDis (ours) | $\mathbf{0.833 \pm 0.068}$ | $\mathbf{0.200 \pm 0.119}$ | $\mathbf{0.065 \pm 0.009}$ | $\mathbf{0.394 \pm 0.132}$ |
| 3D Shapes | | | | |
| VAE | $\mathbf{1.0 \pm 0.0}$ | $0.729 \pm 0.070$ | $0.160 \pm 0.050$ | $0.952 \pm 0.023$ |
| VAE + TopDis (ours) | $\mathbf{1.0 \pm 0.0}$ | $\mathbf{0.835 \pm 0.012}$ | $\mathbf{0.216 \pm 0.020}$ | $\mathbf{0.977 \pm 0.023}$ |
| 3D Faces | | | | |
| VAE | $0.96 \pm 0.03$ | $0.525 \pm 0.051$ | $0.059 \pm 0.013$ | $0.813 \pm 0.063$ |
| VAE + TopDis (ours) | $\mathbf{1.0 \pm 0.0}$ | $\mathbf{0.539 \pm 0.037}$ | $\mathbf{0.063 \pm 0.011}$ | $\mathbf{0.831 \pm 0.023}$ |

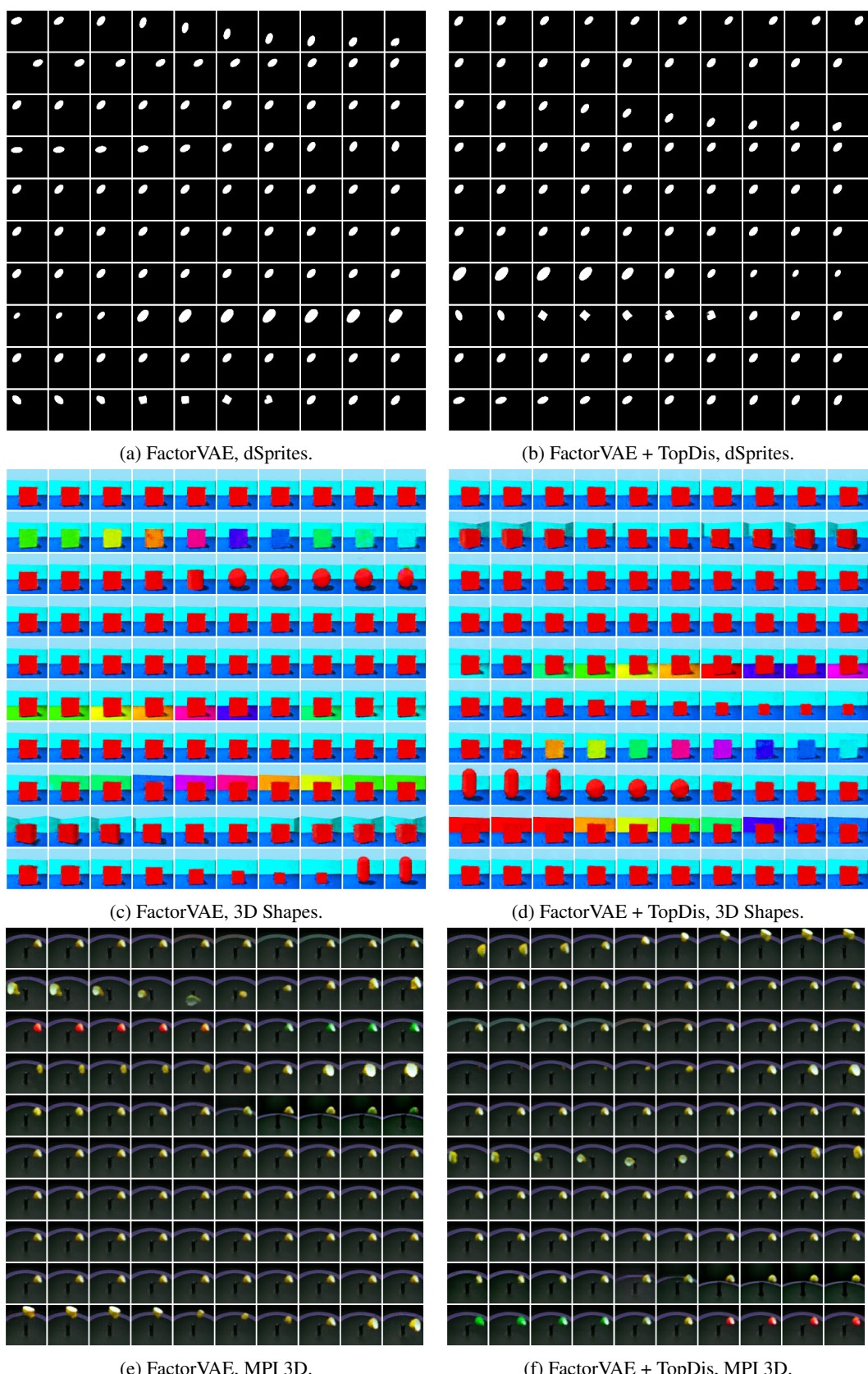

(a) FactorVAE, dSprites.      (b) FactorVAE + TopDis, dSprites.

(c) FactorVAE, 3D Shapes.      (d) FactorVAE + TopDis, 3D Shapes.

(e) FactorVAE, MPI 3D.      (f) FactorVAE + TopDis, MPI 3D.

Figure 20: FactorVAE and FactorVAE + TopDis latent traversals .

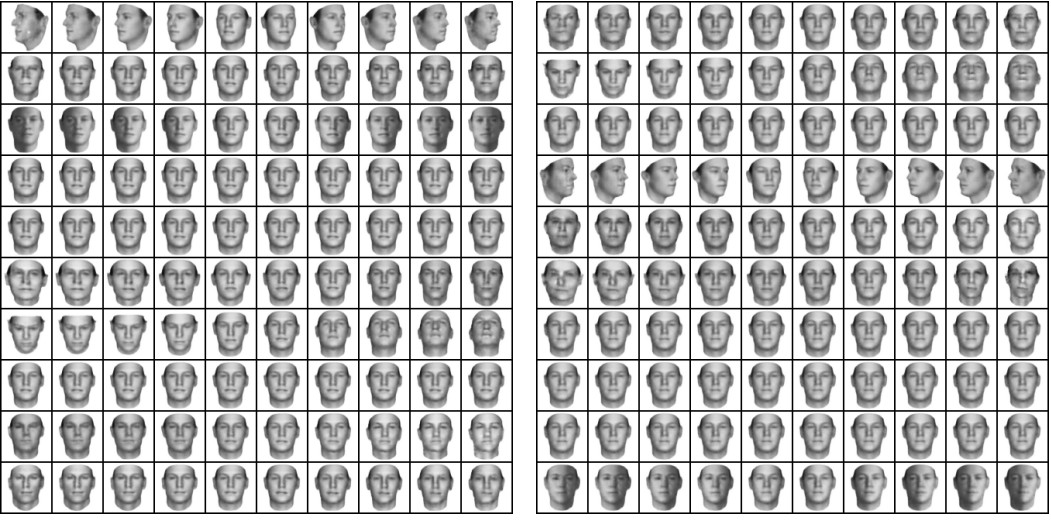

(a) FactorVAE, 3D Faces.

(b) FactorVAE + TopDis, 3D Faces.

Figure 21: FactorVAE and FactorVAE + TopDis latent traversals, 3D Faces.

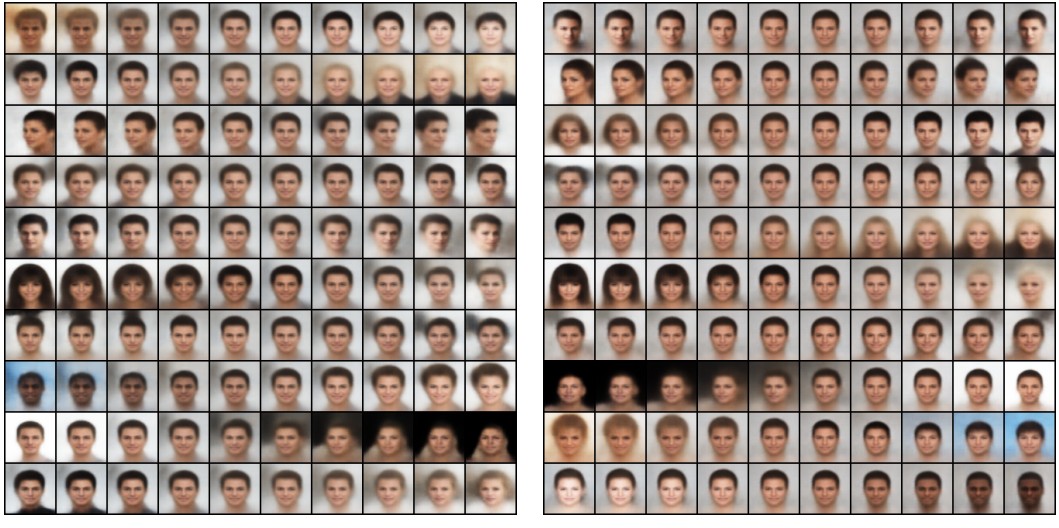

(a) FactorVAE, CelebA.

(b) FactorVAE + TopDis, CelebA.

Figure 22: FactorVAE and FactorVAE + TopDis latent traversals, CelebA.

