# OpenReview forum: "Disentanglement Learning via Topology"
_ICLR.cc/2024/Conference — Submitted to ICLR 2024_

### Official Review · Reviewer_T3Gf · 2023-10-29

**Soundness:** 3 good
**Presentation:** 2 fair
**Contribution:** 2 fair
**Rating:** 5
**Confidence:** 3

**Summary:**

In this paper, the authors propose a method, named TopDis (Topological Disentanglement), for learning disentangled representations via adding a multi-scale topological loss term. The experiments results show that the proposed TopDis loss improves disentanglement scores such as MIG, FactorVAE score, SAP score and DCI disentanglement score with respect to state-of-the-art results while preserving the reconstruction quality.

**Strengths:**

- This paper is the first to introduce the use of a topological regularization term in the field of disentangled representation learning.
- The topological regularization term is shown to be effective across multiple VAE models and metrics.
- The regularization term proposed in this paper is also demonstrated to be effective for discovering pre-trained StyleGAN models.

**Weaknesses:**

- The paper lacks a clear reasonable explanation as to why topological constraints are meaningful/effective for disentanglement representation learning.
- The new loss function was already proposed in a 2022 ICML paper [a]. The main contribution of this work is applying it to disentanglement, making the explanation of the above issue crucial for this paper.
- The experiments focus on models with some disentanglement capabilities, but the effectiveness of this regularization term on vanilla VAEs has not been studied.
- The performance of vanilla VAEs presented in this paper show high DCI performance, but other papers [b] report poor performance instead. A reasonable explanation is needed, and it would be helpful to include evaluation code in the supplementary materials.

[a] Representation Topology Divergence: A method for comparing neural network representations.

[b] β-VAE: LEARNING BASIC VISUAL CONCEPTS WITH A CONSTRAINED VARIATIONAL FRAMEWORK

[c] Disentangling by Factorising

**Questions:**

See weakness

---

> ### Author Response · Authors · 2023-11-23
> **Response to the Review T3Gf #1**
>
> Thank you for your time and the thorough review. We will improve the presentation according to the suggestions. Below we address specific concerns one by one.
>
> __W1__: _The paper lacks a clear reasonable explanation as to why topological constraints are meaningful/effective for disentanglement representation learning._
>
> __A__: Thank you for the feedback. We are adding a clarification deducing the smallness of RTD from the  disentanglement conditions and topological properties of symmetry action.
> Here is the principal explanation.  The Lie group(oid) symmetry $G$ action on the support of data distribution is continuous and invertible. This implies that for any subset of the support of data distribution, the image of the subset under symmetry  $g\in G$ has the same homology or the same group of topological features. The preservation of topological features at multiple scales can be tested with the help of the representation topology divergence (RTD). If RTD is small between a sample from $X$ and its symmetry shift, then the groups of topological features at multiple scales under such symmetry $G$ action are preserved.
>
> Also the smallness of RTD implies the smallness of the disentanglement topological measure from (Zhou et al 2021) based on the geometry scores of data subsets conditioned to a fixed value of a latent code. Such subsets for different fixed values of the latent code are also related via the symmetry shift action, and if RTD between them is small, then the distance between their persistence diagrams and hence the metric from loc cit is small as well.
>
> __W2__: _The new loss function was already proposed in a 2022 ICML paper [a]. The main contribution of this work is applying it to disentanglement, making the explanation of the above issue crucial for this paper._
>
> __A__: Thank you for the remark. As we mentioned in Appendix, the paper [a]  describes briefly an application  of topological metric to evaluation of interpretable directions in a simple synthetic dataset. They compare topological dissimilarity in data submanifolds corresponding to slices in the latent space conditioned by a fixed latent code value.
> In our work, we propose the new differentiable loss function TopDis (Eqn. 3), which measures the topological discrepancy, as evaluated by RTD, between an arbitrary data sample from decoder and its symmetry shift obtained via group(oid) action shifts preserving the Gaussian distribution.
>
> __W3__: _The experiments focus on models with some disentanglement capabilities, but the effectiveness of this regularization term on vanilla VAEs has not been studied._
>
> __A__: Thank you for the valuable feedback. Here are the results for VAE+TopDis.
> As these results demonstrate,  VAE+TopDis outperformed VAE as measured by the 4 disentanglement metrics.
>
> | Method                                |    FactorVAE score |  MIG              |  SAP                  | DCI, dis.            |
> |--------------------------------------|--------------------------|-------------------|-----------------------|----------------------|
> | dSprites | | | | |
> | VAE                           | 0.781 ± 0.016 | 0.170 ± 0.072 | 0.057 ± 0.039 | 0.314 ± 0.072 |
> | VAE + TopDis (ours) | **0.833 ± 0.068**  | **0.200 ± 0.119**  | **0.065 ± 0.009**  | **0.394 ± 0.132** |
> | 3D Shapes | | | | |
> | VAE                 | 1.0 ± 0.0 | 0.729 ± 0.070 | 0.160 ± 0.050 | 0.952 ± 0.023 |
> | VAE + TopDis (ours) | **1.0 ± 0.0** | **0.835 ± 0.012** | **0.216 ± 0.020** | **0.977 ± 0.023** |
> | 3D Faces | | | | |
> | VAE                 | 0.96 ± 0.03 | 0.525 ± 0.051 | 0.059 ± 0.013 | 0.813 ± 0.063 |
> | VAE + TopDis (ours) | **1.0 ± 0.0** | **0.539 ± 0.037** | **0.063 ± 0.011** | **0.831 ± 0.023** |
>
> We had also described the experiments for VAE vs VAE+TopDis-C models in Appendix in Table 6.
>
> (cont'd below)

---

> > ### Author Response · Authors · 2023-11-23
> > **Response to the Review T3Gf #2**
> >
> > __W4__: _The performance of vanilla VAEs presented in this paper show high DCI performance, but other papers [b] report poor performance instead. A reasonable explanation is needed, and it would be helpful to include evaluation code in the supplementary materials._
> >
> > __A__: Thanks for the constructive comment. Our results are somewhat consistent with the literature: the superiority of Vanilla VAE w.r.t beta-VAE for MPI3D was also observed by Locatello et al, 2020 (see Figure 7, bottom row). In [b], beta-VAE performed better than VAE on dSprites which is also consistent with our results.
> > We note that our experimental setup is different from what is used in some other works. In particular, we train the models for 1 million iterations compared to 300 000 iterations used in other works.  We are going to double check experimental results concerning vanilla VAE on MPI3D.
> >
> > We would like to emphasize that our main objective was to improve the SOTA methods, and that is what we achieved. The TopDis loss have consistently improved the performance of β-VAE, β-TCVAE, ControlVAE, FactorVAE and DAVA models in terms of disentanglement scores (MIG, FactorVAE score, SAP score, DCI disentanglement score) while preserving the reconstruction quality.
> >
> >  _Concluding remarks_. Please respond to our post to let us know if the clarifications above suitably address your concerns about our work. We are happy to address any remaining points during the discussion phase; if the responses above are sufficient, we kindly ask that you consider raising your score.

---

> > > ### Comment · Reviewer_T3Gf · 2023-11-23
> > >
> > > Thank you for your response, some of the concerns are addressed. However, I still have the following questions:
> > > 1. Eqn. 3 is defined as the RTD between the sampled images between the original data and the shifted data, what is the difference between RTD in this paper and in [1]?
> > > 2. The DCI performance of the vanilla VAE on 3D Shapes is very high (0.95), which outperforms most of the method proposed recently. I have concern in the evaluation of DCI metric.
> > > 3. Does the clarification:
> > > > A: Thank you for the feedback. We are adding a clarification deducing the smallness of RTD from the disentanglement co...
> > >
> > > add in the main paper? I think it is important to add this part into the main paper.

---

> ### Author Response · Authors · 2023-11-23
> **Response to the Review T3Gf**
>
> Thank you for your valuable feedback.
>
> 1. In [1], the topological metrics is measured between samples whose latent codes necessarily belong to a hyperplane conditioned on the fixed value of one of $z_i$. During the training, the latent codes of data points obtained from the encoder never lie on a hyperplane, so this approach is not applicable for the optimization of the topological loss term. Instead, we work with an arbitrary data points sample obtained from the encoder on which acts the symmetry group(oid) shift.
> 2. For DCI calculation, we used the evaluation code from the commonly used disentanglement lib: https://github.com/google-research/disentanglement_lib .
> We note that we trained the models for 1 million iterations compared to 300k iterations used in several other works which can be a possible source of disagreement.
> Also, in most recent papers only more advanced methods were compared but not the vanilla VAE baseline.
> 3. We have added this and other improvements to the main text, the additions are highlighted by yellow in the revision. We are also adding, as a byproduct, the proposition clarifying the relation between the two approaches to the definition of disentanglement, the factor-independence based and the symmetry based.
>
>  _Concluding remark_. Please respond to our post to let us know if the clarifications above suitably address your concerns about our work. We are happy to address any remaining points; if the responses above are sufficient, we kindly ask that you consider raising your score.

---

### Official Review · Reviewer_2nX5 · 2023-10-30

**Soundness:** 3 good
**Presentation:** 2 fair
**Contribution:** 3 good
**Rating:** 5
**Confidence:** 3

**Summary:**

This paper proposed a novel Topological Disentanglement loss (TopDis loss) that can be added to any VAE-type loss to improve the disentanglement by encouraging the preservation of topological similarity in the generated samples with shifted latent space. Experiments demonstrated the proposed TopDis loss increases the disentanglement performance of several SOTA methods for various disentanglement metrics and datasets.

**Strengths:**

(1) Inspired by [1], the proposed differentiable Representation Topology Divergence (RTD) as a loss for the VAE-framework looks promising to improve the disentanglement.

(2) Rich experiments are conducted to evaluate the performance of the proposed TopDis loss for various VAE-based methods.

[1] Barannikov, Serguei, et al. "Representation topology divergence: A method for comparing neural network representations." ICML 2022.

**Weaknesses:**

(1) It is unclear how the hyper-parameters in Eqn (4) affect the performance. There are γ_1 and γ_2 in Table 9 (appendix N), but there is only one γ in Eqn (4).

(2) In Table 1, it seems that some advanced disentanglement methods performed significantly worse than the vanilla VAE (e.g. FactorVAE on 3dshapes, and β-TCVAE on MPI3D, etc), making it a little suspicious for the experimental results and/or the model selections of baselines. Besides, two important evaluations of VAE+TopDis and β-TCVAE+TopDis are missing.

(3) The evaluation of how the proposed methods handle the tradeoff between disentanglement and reconstruction is limited. Besides Table 4 and Table 8, the authors are encouraged to report the reconstruction errors of the proposed method with and without "gradient orthogonalization" for a complete comparison with the baselines. Did the "gradient orthogonalization" apply to the baselines as well?

**Questions:**

(1) The authors are encouraged to respond to the concerns above.

(2) How the γ should be selected for different VAE-based methods? Does TopDis improve disentanglement when β is already very large? How does the TopDis loss affect the optimization of the original disentanglement loss in those baselines (like the total correction in TC-VAE and FactorVAE)?

---

> ### Author Response · Authors · 2023-11-22
> **Response to the Reviewer 2nX5, #1**
>
> __W1__: _It is unclear how the hyper-parameters in Eqn (4) affect the performance. There are γ_1 and γ_2 in Table 9 (appendix N), but there is only one γ in Eqn (4)._
>
> __A__: Thank you for your remark. The sensitivity w.r.t. to $\gamma$ in Eqn (4) is provided in Appendix Q.
> In Appendix Q, $\gamma_1$ denotes the weight for Total Correlation loss from the FactorVAE model while $\gamma_2$ denotes the weight for TopDis loss from the equation (4). We have added the necessary clarification to the Appendix Q and renamed $\gamma_1$ to $\gamma_{TC}$ (stands for Total Correlation) and $\gamma_2$ to $\gamma_{TD}$ (stands for TopDis) as more suitable ones.
>
> __W2__: _In Table 1, vanilla VAE performed better than some other methods (e.g. FactorVAE on 3d Shapes, and β-TCVAE on MPI3D, etc), evaluations of VAE+TopDis and β-TCVAE+TopDis are not included in Table 1._
>
> __A__:  Thank you for your remarks.
> 1. We provide evaluations of β-TCVAE+TopDis and we have updated Table 1 accordingly.
> Addition of TopDis improves β-TCVAE as measured by the 4 disentanglement metrics.
>
> | Method                                |    FactorVAE score |  MIG              |  SAP                  | DCI, dis.            |
> |--------------------------------------|--------------------------|-------------------|-----------------------|----------------------|
> | dSprites | | | | |
> | β-TCVAE                           | 0.810 ± 0.058 | 0.332 ± 0.029 | 0.045 ± 0.004 | 0.543 ± 0.049 |
> | β-TCVAE + TopDis (ours) | **0.821 ± 0.034**  | **0.341 ± 0.021** | **0.051 ± 0.004** | **0.556 ± 0.042** |
> | 3D shapes | | | | |
> | β-TCVAE                          | 0.909 ± 0.079 | 0.693 ± 0.053 | 0.113 ± 0.070  | 0.877 ± 0.018 |
> | β-TCVAE + TopDis (ours) | **1.0 ± 0.0**   | **0.751 ± 0.051**  | **0.147 ± 0.064** | **0.901 ± 0.014** |
> | 3D Faces | | | | |
> | β-TCVAE                           | 1.0 ± 0.0 | 0.568 ± 0.063 | 0.060 ± 0.017 | 0.822 ± 0.033 |
> | β-TCVAE + TopDis (ours) | 1.0 ± 0.0 | **0.591 ± 0.058**  | **0.062 ± 0.011** | **0.859 ± 0.031** |
> | MPI 3D | | | | |
> | β-TCVAE                           | 0.365 ± 0.042 | 0.174 ± 0.018  | 0.080 ± 0.013 | 0.225 ± 0.061 |
> | β-TCVAE + TopDis (ours) | **0.496 ± 0.039**  | **0.280 ± 0.013** | **0.143 ± 0.009** | **0.340 ± 0.055** |
>
> 2. Also, we are running experiments with VAE+TopDis, please find the results below.
> We see that VAE+TopDis outperformed VAE as measured by the 4 disentanglement metrics.
>
> | Method                                |    FactorVAE score |  MIG              |  SAP                  | DCI, dis.            |
> |--------------------------------------|--------------------------|-------------------|-----------------------|----------------------|
> | dSprites | | | | |
> | VAE                           | 0.781 ± 0.016 | 0.170 ± 0.072 | 0.057 ± 0.039 | 0.314 ± 0.072 |
> | VAE + TopDis (ours) | **0.833 ± 0.068**  | **0.200 ± 0.119**  | **0.065 ± 0.009**  | **0.394 ± 0.132** |
> | 3D Shapes | | | | |
> | VAE                 | 1.0 ± 0.0 | 0.729 ± 0.070 | 0.160 ± 0.050 | 0.952 ± 0.023 |
> | VAE + TopDis (ours) | **1.0 ± 0.0** | **0.835 ± 0.012** | **0.216 ± 0.020** | **0.977 ± 0.023** |
> | 3D Faces | | | | |
> | VAE                 | 0.96 ± 0.03 | 0.525 ± 0.051 | 0.059 ± 0.013 | 0.813 ± 0.063 |
> | VAE + TopDis (ours) | **1.0 ± 0.0** | **0.539 ± 0.037** | **0.063 ± 0.011** | **0.831 ± 0.023** |
>
> 3. Indeed, vanilla VAE performed better than some other methods in some cases. This is consistent with actual papers: superiority of Vanilla VAE w.r.t beta-VAE for MPI3D was also observed by Locatello et al, 2020 (see Figure 7, bottom row). We have decided to double-check them and will report results in Appendix, as in most recent papers only more advanced methods were used but not the vanilla VAE baseline. Also we trained the models for 1 million iterations compared to 300 000 iterations used in several other works which can be a possible source of disagreement.
>
> We would like to emphasize that our main objective was to improve the SOTA methods, and that is what we achieved. The TopDis loss have consistently improved the performance of β-VAE, β-TCVAE, ControlVAE, FactorVAE and DAVA models in terms of disentanglement scores (MIG, FactorVAE score, SAP score, DCI disentanglement score) while preserving the reconstruction quality.

---

> ### Author Response · Authors · 2023-11-22
> **Response to the Reviewer 2nX5, #2**
>
> __W3__: _The evaluation of how the proposed methods handle the tradeoff between disentanglement and reconstruction is limited. Besides Table 4 and Table 8, the authors are encouraged to report the reconstruction errors of the proposed method with and without "gradient orthogonalization" for a complete comparison with the baselines. Did the "gradient orthogonalization" apply to the baselines as well?_
>
> __A__: Thank you for your question and remark. Figure 17 in Appendix illustrates the positive effect of gradient orthogonalization on reconstruction error during training. We do not apply gradient orthogonalization in each experiment but only when the proposed loss can slightly hinder the reconstruction objective.. In our experiments, we orthogonalize the gradient of the proposed TopDis loss w.r.t. the reconstruction loss. Since there is a known tradeoff between reconstruction quality and disentanglement, our primary goal is to develop an additional loss that would enhance the underlying base model but would not result in increase of reconstruction error. While in our experiments we do not apply gradient orthogonalization separately for the baseline models, it is possible to apply the gradient orthogonalization of disentanglement-promoting loss (e.g. Total Correlation for FactorVAE, additional KL-divergence for BetaVAE, etc.) for the baseline model. While a similar technique is known, to the best of our knowledge, it has not been used in the context of learning disentangled representations before.
>
> __Q2__: _How the γ should be selected for different VAE-based methods? Does TopDis improve disentanglement when β is already very large?_
>
> __A__: Thank you for your questions.
> 1. In our experiments, we used the following greedy procedure to tune the weight $\gamma$ of the TopDis loss. First, we tune hyperparameters of baseline approaches (i.e. beta-VAE, FactorVAE, ControlVAE, etc.), by selecting ones having both high disentanglement quality and reasonable reconstruction error. A minor exception is the DAVA model, where the hyperparameters are tuned adaptively during training. Then, we select the best $\gamma$ from the range [1, 6] by the disentanglement quality. We observe that optimal $\gamma$ often coincide for different datasets.
>
> 2. High values of $\beta$ typically lead to high reconstruction error, we do not explore this setup in our experiments. Our goal is to enhance the underlying model to provide both good disentanglement and reconstruction quality.
>
> _Concluding remarks_. Please respond to our post to let us know if the clarifications above suitably address your concerns about our work. We are happy to address any remaining points during the discussion phase; if the responses above are sufficient, we kindly ask that you consider raising your score.

---

### Official Review · Reviewer_VGnt · 2023-10-31

**Soundness:** 2 fair
**Presentation:** 2 fair
**Contribution:** 2 fair
**Rating:** 5
**Confidence:** 4

**Summary:**

The paper proposed a disentanglement regularization term based on topology, to constrain the manifold relation between the latent points of original images and shifted images. The authors provided extensive experiments on VAE-based methods and showed the effectiveness of the proposed methods.

**Strengths:**

1.	It is important to explore the constrain in the manifold of latent space for disentanglement, due to the statistical arguments of Locatello et al. (2019). The paper explored a way from topology and proposed a regularization term, which can be easily optimized.

2.	The paper provided a good formulation of the TopDis loss and how to optimize it in the VAE framework.

**Weaknesses:**

1.	The relation between the constrain on latent space and disentanglement is still unclear, the TopDis is based on VAE-framework, which is based on Probability, and the paper referred to the definition of disentanglement based Group. And the paper failed to connect the above two framework, and making the proposed TopDis only kind of an intuitive necessary condition, as shown in Figure 3.

2.	From Appendix L, the best performance hyperparameters are quite different across different methods and different datasets, is there any guidance or criterion to choose the hyper-parameter?

**Questions:**

1.	My main concern is the relation between the proposed TopDis and disentanglement, is there any theoretical guarantee or deduction?
2.	The authors applied the proposed TopDis to infer disentangled directions in a pretrained style-GAN, is there some quantitative results? Then dose the method can be applied to other disentangled methods?

---

> ### Author Response · Authors · 2023-11-23
> **Response to the Reviewer VGnt, #1**
>
> Thank you for your time and thorough review. We will improve the presentation according to the suggestions. Below we address specific concerns one by one.
>
>
> __W1__: _The relation between the constrain on latent space and disentanglement is still unclear, the TopDis is based on VAE-framework, which is based on Probability, and the paper referred to the definition of disentanglement based Group. And the paper failed to connect the above two framework,and making the proposed TopDis only kind of an intuitive necessary condition,as shown in Figure3._
> __A__: Thank you for the valuable feedback.
> 1.We are adding a clarification on the relation of the probability based definition of disentanglement and the group based definition. Let $q(z)$ be the  aggregate posterior distribution over a latent space, aggregated over the whole dataset $X$. And let $q(z_i)$ be the similar aggregate distribution over the latent code $z_i$. The formula (1) is  valid and defines symmetry group(oid) shifts if we replace the standard normal distribution by any distribution over the real line, we use it with the distribution $q(z_i)$ over the $i-$th latent codes.
>
> _Proposition._ a) If the distribution $q(z)$ is factorized into product $q(z)=\prod_i q(z_i)$, then the shift defined by the formula (1) and acting on a single latent code $z_i$ and leaving other latent codes fixed,  preserves the latent space distribution $q(z)$. This defines the $G_i$ groupoid action on $z$ for any $i$, whose action is then extended to points of the initial dataset $X$ with the help of the decoder-encoder.  b) Conversely, if $q(z)$ is preserved for any $i$ by the shifts acting on $z_i$ and defined via formula (1) from the distribution $q(z_i),$ then $q(z)=\prod_i q(z_i)$.
> _Proof._ a) The shift defined by (1) for the distribution $q(z_i)$ acting on the latent space, preserves also any $q(z_j)$ for $ j\neq i$. b) The result follows from the case of an arbitrary distribution over a pair of random variables $z_1, z_2$. For two variables, it follows from the Bayes formula that the shifts of $z_1$ preserve the conditional $q(z_2\vert z_1)$. Since the group(oid) action is transitive it follows that the conditional does not depend on $z_1$, and hence $q(z_1,z_2)=q(z_1)q(z_2)$.
>
> To the best of our knowledge, we are the first to impose the preservation of the distributions under the symmetry groupoid action condition in the standard normal distribution variational autoencoder   framework. On the one hand, this implies that such symmetry action is necessarily defined not by a group but by the groupoid and also using the specific shifts as in equation (1). On the other hand, this distribution preservation by groupoid is the main ingredient making the two definitions of disentanglement compatible.
> We are adding this result clarifying the correspondence between the two definitions to the paper.
>
> 2.We are adding a clarification deducing the smallness of RTD from the disentanglement conditions and properties of symmetry action on manifolds.Briefly, the Lie group(oid) symmetry $g$ action on the support of data distribution is continuous and invertible. This implies that for any subset of the support of data distribution, the image of the subset under $g$ has the same homology or the same group of topological features. The preservation of topological features at multiple scales can be tested with the help of the representation topology divergence (RTD). If RTD is small between a sample from $X$ and its symmetry shift, then the groups of topological features at multiple scales are preserved.
>
> Also the smallness of RTD implies the smallness of the disentanglement measure from (Zhou et al. 2020) based on the geometry scores of data subsets conditioned to a fixed value of a latent code. Such subsets for different fixed values of the latent code are also related via the symmetry shift action, and if RTD between them is small, the distance between their persistence diagrams and hence the metric from loc cit is small as well.
>
> __W2__: _From Appendix L, the best performance hyperparameters are quite different across different methods and different datasets, is there any guidance or criterion to choose the hyper-parameter?_
>
> __A__:  In our experiments, we used the following greedy procedure to tune the weight $\gamma$ of the TopDis loss. First, we tune hyperparameters of baseline approaches (i.e. beta-VAE, FactorVAE, ControlVAE, etc.), by selecting ones having the high disentanglement quality while keeping reasonable reconstruction error. A minor exception is the DAVA model, where the hyperparameters are tuned adaptively during training. Then, we select the best $\gamma$ weight for TopDis from the range [1, 6] by the disentanglement quality. We observe that optimal $\gamma$ often coincide for different datasets.
>
> (cont'd below)

---

> ### Author Response · Authors · 2023-11-23
> **Response to the Reviewer VGnt, #2**
>
> __Q1__: _My main concern is the relation between the proposed TopDis and disentanglement, is there any theoretical guarantee or deduction?_
>
> __A__: Thank you for the question. (See the answer to W1.)  Firstly, we add the proposition establishing the relation of the probability based definition of disentanglement and the group based definition. Secondly, we add the clarification deducing the TopDis minimization from the disentanglement conditions and properties of symmetry groupoid action, see the answer to W1.
>
> __Q2__: _The authors applied the proposed TopDis to infer disentangled directions in a pretrained style-GAN, is there some quantitative results? Then dose the method can be applied to other disentangled methods?_
>
> __A__: Comparison of methods dedicated to the unsupervised discovery of disentangled directions in StyleGAN is qualitative since the FFHQ dataset doesn't have labels. Our goal is to demonstrate the applicability of the TopDis loss for this problem. See section 5.3 for details.
>
>  _Concluding remarks_. Please respond to our post to let us know if the clarifications above suitably address your concerns about our work. We are happy to address any remaining points during the discussion phase; if the responses above are sufficient, we kindly ask that you consider raising your score.

---

### Official Review · Reviewer_1sM8 · 2023-10-31

**Soundness:** 3 good
**Presentation:** 3 good
**Contribution:** 2 fair
**Rating:** 6
**Confidence:** 4

**Summary:**

The authors of this paper present TopDis, which is a regularizer based on Representation Topology Divergence (RTD). In this approach, the objective to be optimized is a combination of “classic” VAE loss and TopDis loss. Unlike the preceding approaches, topDis does not assume statistical independence between the factors of variations. Generally, introducing this loss term appears to further improve the current SOTA values for several disentanglement metrics (FactorVAE, MIG, SAP and DCI) across several different datasets (dSprites, 3D Shapes, 3D Faces, MPI 3D).

**Strengths:**

1. The paper is clearly written and easy to follow. In detail:

a. The authors explain the task of disentanglement rather clearly by providing a succinct overview of previous works.

b. The motivation and contribution of the paper are also clearly defined with an intuitive explanation of the designed methodology.

2. The authors provide a variety of experiments and ablations, helping to evaluate their proposed disentanglement regularization loss practically. In detail:

a. The experiments (Table 1) appear comprehensive (except for the vanilla VAE; we will explain in the weakness section our concerns).

b. The authors also provide enough qualitative examples, comparing models trained with TopDis regularizer and without.

c. The architecture is succinctly described in the Appendix

3. Computational complexity is also discussed in the Appendix, which is crucial for ML algorithms nowadays.

**Weaknesses:**

1. One of the contributions the authors mention is: “We improve the reconstruction quality by applying gradient orthogonalization;” - however, this contribution is only briefly mentioned in the conclusion and analyzed in the Appendix in greater detail. We suggest the authors to “move” the gradient orthogonalization part to the main paper.

2. As the authors explained, the RTD was defined in a previous work, but we believe it is important to be defined in the main paper.

3. In section 4.1, bullets (2-4). In (2), g\inG appears to be applied to both pixel and latent space. Later in (3,4), where decomposition G is defined, it seems that it can be applied only in the latent space. We believe the authors should re-write this part, clarifying how G can be applied in the pixel space or, if that is not the case remove from (2) the application of g in the pixel space.

4. In equation (4) regularization parameter /gamma is defined. Later in the appendix Q, \gamma_1, and \gamma_2 are used in the ablation table. Does this correspond, instead, to the loss: \gamma_1 L_{VAE-based} + \gamma_2 L_{TD}.

5. In page 5 footnote, the authors state that RPT can be computed in latent space instead of pixel space. Can the authors provide ablations in the appendix exploring this direction? Do the authors have insights into how this change can affect the final trained model?

6. Finally, our main concern is whether the proposed regularizer contributes to the learning of the disentangled representation or the used base models (i.e., \beta-VAE, Factor-VAE). Since, in the main paper, only the models with already disentanglement remedies are explored and not the vanilla VAE. More concerning in the ablation, VAE+TopDis is explored, but it seems that the training is not the same as the VAE reported in the main paper. Our guess is that the models in the ablation were trained for less number of iterations. We encourage the authors to include in the main paper VAE+TopDis trained under the same conditions (i.e. same number of iterations) as the reported VAE in Table 1. This will help readers understand to what extent the TopDis regularizer helps learn disentangled representations

**Questions:**

See above

---

> ### Author Response · Authors · 2023-11-21
> **Response to the Review 1sM8**
>
> Thank you for your time and thorough review highlighting strengths of our paper. We will improve the presentation according to the suggestions. Below we address specific concerns one by one.
>
> __W1__: _We suggest the authors to “move” the gradient orthogonalization part to the main paper._
> __A__: Thank you for the suggestion, we have moved the details on gradient orthogonalization to the main part.
>
> __W2__: _Move more details on formal definition of RTD to the main paper_
> __A__: Thank you for the suggestion. We are adding more details on the formal definition to the discussion of the RTD definition in Section 3.1.
>
> __W3__: _Clarify how $G$ can be applied in the pixel space._
> __A__: Thank you for your question. The equation $f(g(z))=g(f(z))$ concerns the outcome of the learning process, which consists, in particular, in finding a priori unknown symmetries in the data. Given the decoder map $f$, this equation defines the symmetry action on $X$ preserving the data distribution.
>
> __W4__: _Clarify notations \gamma_1 and \gamma_2 in the ablation table in Appendix Q._
> __A__: Thank you for your remark. In Appendix Q, \gamma_1 denotes the weight for Total Correlation loss from the FactorVAE model while \gamma_2 denotes the weight for TopDis loss from the equation (4). We have added the necessary clarification to the Appendix Q and renamed \gamma_1 to \gamma_TC (stands for Total Correlation) and \gamma_2 to \gamma_TD (stands for TopDis) as more suitable ones.
>
> __W5__: _Can RTD be applied in latent space instead of pixel space?_
> __A__: Thank you for your question. In the footnote on page 5, we mean that it is possible to compute TopDis loss either between the images (i.e. in pixel space) or between their representations (for example, produced by another pretrained model). It can be beneficial for different reasons to compare the multiscale topology of data in a representation space. For example, common disentanglement datasets have typical image resolution 64x64. However, in case of significantly higher image resolution, it could be beneficial to utilize representations instead of images. As a proof of concept, in the subsection 5.3 and Appendix H, we provide the results on unsupervised discovery of disentangled directions in StyleGAN where we compute the RTD in the representation space instead of pixel space. With these experiments, we illustrate that RTD finds meaningful directions using the representation space as well.
>
>
> __W6__: _Clarification on VAE+TopDis_
> __A__: Thank you for the question.  We are now running the experiments for VAE+TopDis.  Here are the results for VAE+TopDis vs VAE :
>
>
> | Method                                |    FactorVAE score |  MIG              |  SAP                  | DCI, dis.            |
> |--------------------------------------|--------------------------|-------------------|-----------------------|----------------------|
> | dSprites | | | | |
> | VAE                           | 0.781 ± 0.016 | 0.170 ± 0.072 | 0.057 ± 0.039 | 0.314 ± 0.072 |
> | VAE + TopDis (ours) | **0.833 ± 0.068**  | **0.200 ± 0.119**  | **0.065 ± 0.009**  | **0.394 ± 0.132** |
> | 3D Shapes | | | | |
> | VAE                 | 1.0 ± 0.0 | 0.729 ± 0.070 | 0.160 ± 0.050 | 0.952 ± 0.023 |
> | VAE + TopDis (ours) | **1.0 ± 0.0** | **0.835 ± 0.012** | **0.216 ± 0.020** | **0.977 ± 0.023** |
> | 3D Faces | | | | |
> | VAE                 | 0.96 ± 0.03 | 0.525 ± 0.051 | 0.059 ± 0.013 | 0.813 ± 0.063 |
> | VAE + TopDis (ours) | **1.0 ± 0.0** | **0.539 ± 0.037** | **0.063 ± 0.011** | **0.831 ± 0.023** |
>
> The reconstruction loss for VAE+TopDis: 9.54±0.19 (dSprites), 3489.533±1.502 (3DShapes), 1376.218±0.316 (3DFaces)
>
> As we see, VAE+TopDis outperformed VAE as measured by the 4 disentanglement metrics.
>
> In the experiments in Table 6 in Appendix G the training procedure was identical to that of Table 1 in the main paper, except that we have used a slightly smaller architecture in Table 6 for dSprites for both VAE and VAE+TopDis-C compared with the experiments from Table 1. For this reason, the metrics values differed in Table 1 and Table 6 for VAE at dSprites dataset. However, we highlight that the experimental setup for 3D Shapes is identical in both Table 1 and Table 6. And that our setup is always consistent when we compare a base model with and without TopDis loss.
>
> Also, our main objective was to improve the SOTA methods, and that is what we achieved. The TopDis loss  have consistently improved the performance of β-VAE, β-TCVAE, ControlVAE, FactorVAE and DAVA models in terms of disentanglement scores (MIG, FactorVAE score, SAP score, DCI disentanglement score) while preserving the reconstruction quality.
>
> _Concluding remarks_. Please respond to our post to let us know if the clarifications above suitably address your concerns about our work. We are happy to address any remaining points during the discussion phase; if the responses above are sufficient, we kindly ask that you consider raising your score.

---

### Author Response · Authors · 2023-11-23
**Summary of Author Response to All the Reviewers**

We appreciate all the Reviewers for their time and valuable feedback.
We have updated the manuscript and highlighted changes with yellow.

Here are some of principal points in our responses:

__Q__: _The effectiveness of the proposed TopDis regularization term on vanilla VAEs has not been studied._

__A__: We carried out additional experiments with VAE+TopDis.
As these results demonstrate,  VAE+TopDis outperformed VAE as measured by the 4 disentanglement metrics.

| Method                                |    FactorVAE score |  MIG              |  SAP                  | DCI, dis.            |
|--------------------------------------|--------------------------|-------------------|-----------------------|----------------------|
| dSprites | | | | |
| VAE                           | 0.781 ± 0.016 | 0.170 ± 0.072 | 0.057 ± 0.039 | 0.314 ± 0.072 |
| VAE + TopDis (ours) | **0.833 ± 0.068**  | **0.200 ± 0.119**  | **0.065 ± 0.009**  | **0.394 ± 0.132** |
| 3D Shapes | | | | |
| VAE                 | 1.0 ± 0.0 | 0.729 ± 0.070 | 0.160 ± 0.050 | 0.952 ± 0.023 |
| VAE + TopDis (ours) | **1.0 ± 0.0** | **0.835 ± 0.012** | **0.216 ± 0.020** | **0.977 ± 0.023** |
| 3D Faces | | | | |
| VAE                 | 0.96 ± 0.03 | 0.525 ± 0.051 | 0.059 ± 0.013 | 0.813 ± 0.063 |
| VAE + TopDis (ours) | **1.0 ± 0.0** | **0.539 ± 0.037** | **0.063 ± 0.011** | **0.831 ± 0.023** |

__Q__: _The effectiveness of the proposed TopDis regularization term on β-TCVAE has not been studied._

__A__: We provide evaluations of β-TCVAE+TopDis and we have updated Table 1 accordingly.
Addition of TopDis improves β-TCVAE as measured by the 4 disentanglement metrics.

| Method                                |    FactorVAE score |  MIG              |  SAP                  | DCI, dis.            |
|--------------------------------------|--------------------------|-------------------|-----------------------|----------------------|
| dSprites | | | | |
| β-TCVAE                           | 0.810 ± 0.058 | 0.332 ± 0.029 | 0.045 ± 0.004 | 0.543 ± 0.049 |
| β-TCVAE + TopDis (ours) | **0.821 ± 0.034**  | **0.341 ± 0.021** | **0.051 ± 0.004** | **0.556 ± 0.042** |
| 3D shapes | | | | |
| β-TCVAE                          | 0.909 ± 0.079 | 0.693 ± 0.053 | 0.113 ± 0.070  | 0.877 ± 0.018 |
| β-TCVAE + TopDis (ours) | **1.0 ± 0.0**   | **0.751 ± 0.051**  | **0.147 ± 0.064** | **0.901 ± 0.014** |
| 3D Faces | | | | |
| β-TCVAE                           | 1.0 ± 0.0 | 0.568 ± 0.063 | 0.060 ± 0.017 | 0.822 ± 0.033 |
| β-TCVAE + TopDis (ours) | 1.0 ± 0.0 | **0.591 ± 0.058**  | **0.062 ± 0.011** | **0.859 ± 0.031** |
| MPI 3D | | | | |
| β-TCVAE                           | 0.365 ± 0.042 | 0.174 ± 0.018  | 0.080 ± 0.013 | 0.225 ± 0.061 |
| β-TCVAE + TopDis (ours) | **0.496 ± 0.039**  | **0.280 ± 0.013** | **0.143 ± 0.009** | **0.340 ± 0.055** |

__Q__: _High performance of vanilla VAE in some cases._

__A__: Indeed, vanilla VAE performed better than some other methods in some cases. This is consistent with actual papers: superiority of Vanilla VAE w.r.t beta-VAE for MPI3D was also observed by Locatello et al, 2020 (see Figure 7, bottom row). We note that we trained the models for 1 million iterations compared to 300k iterations used in several other works which can be a possible source of disagreement. Also, in most recent papers only more advanced methods were compared but not the vanilla VAE baseline. For evaluation of disentanglement metrics, we used the code from the commonly used disentanglement lib: https://github.com/google-research/disentanglement_lib

---

### Meta-Review · Area_Chair_uqVP · 2023-12-07

**Metareview:**

This paper introduced a differentiable topological loss for disentanglement learning. Contrary to the prior works that directly encourage the factorization of the latent variable based on VAE, the authors employed topological similarity on the data manifold as a measurement for disentanglement. This is achieved by measuring RTD (Barannikov et al., 2022) traversing along the latent space, which is added as a regularization to the existing VAE objective.

Four reviewers recommended borderline scores of one borderline accept and three borderline reject. The primary concerns raised by the reviewers were about (1) clarifying the relation between the topological distance and disentanglement of latent factors, (2) clarifying the novelty over Barannikov et al., 2022, and (3) missing ablation studies and comparisons to additional baselines. The authors adequately addressed some of the concerns, but three reviewers maintained their negative recommendations mainly due to insufficient justifications on how RTD contributes to disentanglement learning.

After reading the paper, reviews, and rebuttal, the AC agrees with the reviewers that the relation between RTD and disentanglement learning should be more clearly and thoroughly justified. The rebuttal provided informative clarifications, yet it is still insufficient to understand how the disentanglement imposed by RTD is related to other methods that directly encourage dimension-wise factorization i.e., it is not clear what complementary benefits are provided, since the authors’ justification implies that the same factorization is expected. Such a lack of insights weakens the significance of the new results (i.e., results with different disentangled VAE baselines). For these reasons, AC believes that the paper presents an interesting and promising idea of new loss for disentanglement learning, yet the paper requires substantial revision to provide clear insights on how it works and is complementary to existing works. Hence, AC recommends rejection this time.

**Justification For Why Not Higher Score:**

Insufficient justifications on how RTD contributes to disentanglement learning.

**Justification For Why Not Lower Score:**

N/A

---

### Decision · Program_Chairs · 2024-01-16

Reject